# A simplified method for the detection of convection using high resolution imagery from GOES-16

Yoonjin Lee[1], Christian D. Kummerow[1,2], and Milija Zupanski[2]

[1]Department of Atmospheric Science, Colorado state university, Fort Collins, CO 80521, USA
[2]Cooperative Institute for Research in the Atmosphere, Colorado state university, Fort Collins, CO 80521, USA

*Correspondence to*: Yoonjin Lee (ylee@atmos.colostate.edu)

**Abstract.** The ability to detect convective regions and to add latent heating to drive convection is one of the most important additions to short term forecast models such as National Oceanic and Atmospheric Administration's (NOAA's) High Resolution Rapid Refresh (HRRR) model. Since radars are most directly related to precipitation and are available in high temporal
resolution, their data are often used for both detecting convection and estimating latent heating. However, radar data are limited to land areas, largely in developed nations, and early convection is not detectable from radars until drops become large enough to produce significant echoes. Visible and Infrared sensors on a geostationary satellite can provide data that are more sensitive to small droplets, but they also have shortcomings: their information is almost exclusively from the cloud top. Relatively new geostationary satellites, Geostationary Operational Environmental Satellites-16 and -17 (GOES-16 and GOES-17), along with
Himawari-8, can make up for this lack of vertical information through the use of very high spatial and temporal resolutions, allowing to better observe bubbling features on convective cloud tops. This study develops two algorithms to detect convection at vertically growing clouds and mature convective clouds using 1-minute GOES-16 Advanced Baseline Imager (ABI) data. Two case studies are used to explain the two methods, followed by results applied to one month of data over the contiguous United States. Vertically growing clouds in early stages are detected using decreases in brightness temperatures over ten minutes. For
mature convective clouds which no longer show much decreases in brightness temperature, the lumpy texture from rapid development can be observed using 1-minute high spatial resolution reflectance data. Detection skill of the two methods are validated against Multi-Radar/Multi-Sensor System (MRMS), a ground-based radar product. With the contingency table, results applying both methods to one month data show a relatively low false alarm rate of 14.4% but missed 54.7% of convective clouds detected by the radar product. These convective clouds are largely under optically thick cloud shields, and thus missed from
analysing lumpy textures.

## 1 Introduction

While weather forecast models have improved tremendously throughout the decades (Bauer et al., 2015), local scale phenomena such as convection remain challenging (Yano et al., 2018). Precipitation is especially hard to predict as numerical models struggle with initiating convection in the right location and intensity. To address this issue in short term predictions, many models
now assimilate all-sky radiances and precipitation-related products where available (Benjamin et al., 2016; Bonavita et al., 2017; Geer et al., 2017; Gustafsson et al., 2017; Jones et al., 2016; Migliorini et al., 2018; Scheck et al., 2020). In some forecast models such as the High Resolution Rapid Refresh (HRRR) model in the United States, latent heating is added, along with precipitation affected radiances, to adjust model dynamics to correspond to the observed convection (Benjamin et al., 2016). Latent heating is only added in convective regions because local scale phenomena tend to develop first by convective clouds before detraining
stratiform precipitation. In order to correctly detect convective regions and add heating as accurately as possible, ground-based radars have been used during the short-term forecast. However, ground-based radar data are not available over ocean or

mountainous regions. Therefore, this study explores whether high temporal resolution data from recent operational geostationary satellite, Geostationary Operational Environmental Satellites (GOES) – R Series, can provide similar information as radar for the location of convection so that it can be used for initializing forecast models over regions without ground-based radar.


Convection is classically defined from in-cloud vertical air motions (Steiner et al., 1995). However, since vertical velocity is rarely measured directly, the radar community initially adopted radar reflectivity thresholds to define convection and distinguish it from stratiform precipitation (Churchill and Houze, 1984; Steiner et al., 1995). One problem with using reflectivity threshold is its sensitivity to the selected threshold for convection. If the threshold is set high, convective regions where precipitation has just begun are not captured, while a threshold that is set too low will misclassify some stratiform regions as convective. To address this issue, Churchill and Houze (1984) separated precipitation types by using the horizontal structure of precipitation fields (Steiner et al., 1995). They classified a grid point as convective if the grid point had rain rates twice as high as the average taken over surrounding grid points or had reflectivity over 40dBZ ($\sim 20$ mm h$^{-1}$). Steiner et al. (1995) refined this method with three criteria: intensity, peakedness, and surrounding area. They used the same threshold of 40dBZ for intensity in the first step, but grid points with reflectivity greater than the average reflectivity within a radius of 11km as well as surrounding grid points are also classified as convective. Nonetheless, stratiform regions sometimes can have reflectivity values greater than 40dBZ. Zhang et al. (2008) used two reflectivity criteria for convective precipitation-namely that the reflectivity be greater than 50dBZ at any height and greater than 30dBZ at -10°C height or above. Zhang and Qi (2010) defines a grid point as convective if the vertically integrated liquid water exceeds a threshold of 6.5kg m$^{-2}$. Qi et al. (2013) developed a new algorithm that combined two previous methods from Zhang et al. (2008) and Zhang and Qi (2010). By combining these two methods and modifying the thresholds, they were able to decrease misclassification of stratiform regions with strong bright band features, but could still miss some convective regions in their initial stage due to a high reflectivity threshold. The HRRR model uses a much lower reflectivity threshold of 28dBZ to detect convective regions and assigns a heating increment (Weygandt et al., 2016). While this is significantly lower than the thresholds discussed above, its primary purpose is to initiate convection where there is significant echo present, while relying on the model physics to assign the proper precipitation type.

While radars have been the preferred method for detecting convection, they are not the only instruments available. Visible (VIS) and infrared (IR) radiances also contain some information, although largely limited to cloud top properties. Convection detection algorithms using VIS and IR sensors exist for both convective initiation (CI) and mature stages. With the recent growing interest in machine learning techniques, many studies have applied machine learning methods in detecting convection (Han et al., 2019; Zhang et al., 2019; Cintineo et al., 2020), but knowledge in physical features of convective clouds is still required to construct a model that correctly learns during training. At the initial stages of convection, cloud tops grow vertically, and decrease in brightness temperature ($T_b$) is observed accordingly. Many algorithms use decreased cloud top temperature from the growth (related to the in-cloud vertical velocity) to detect convective regions from various geostationary satellites over the globe such as GOES (Sieglaff et al., 2011; Mecikalski and Bedka, 2006), Himawari-8 (Lee et al., 2017), and Meteosat (Autonès and Moisselin, 2010). Temporal trends of $T_b$ are evaluated on several channels around water vapor absorption band or longwave infrared window band and combinations of these channels. Interest fields for CI include temporal trend of $T_b$ at 10.7μm (or 11.2μm) to infer cloud top cooling rates, (3.9μm – 10.7μm) to infer changes in cloud top microphysics, and (6.5μm – 10.7μm) to infer cloud height changes relative to the tropopause (Mecikalski and Bedka, 2006). Main differences between the algorithms are tracking method of a cloud and time period used to calculate $T_b$ change of the cloud. Clouds are usually tracked with atmospheric motion vectors or a simple overlap method, and temporal trends of $T_b$ are calculated over 15 minutes.

Convective clouds in their mature stage cannot be detected by the abovementioned algorithms as their cloud tops do not grow much in the vertical, and $T_b$ decrease is not a main feature that is applicable to such clouds. Overshooting Top (OT) is one of the clear indications of mature convective clouds, and many existing algorithms used OT feature in such clouds. There are two common approaches to detect OTs: the brightness temperature difference method and the infrared window-texture method (Ai et al., 2017). The brightness temperature difference method uses a difference in $T_b$ between the water vapor (WV) channel and IR window channel ($T_{b,wv} - T_{b,IR}$). Positive values of $T_{b,wv} - T_{b,IR}$ due to the forcing of warm WV from below into the lower stratosphere are used as an indicator of OTs (Setvak et al., 2007). However, since the threshold for the difference between two channels can depend on several factors, Bedka et al. (2010) suggested another method to detect OTs which is called the Infrared window-texture method. This method takes advantage of a feature of OT in that it is an isolated region with cold $T_b$ surrounded by relatively warm anvil region (Bedka et al., 2010). This method, unfortunately, cannot avoid having to choose $T_b$ thresholds that vary according to seasons or regions (Dworak et al., 2012). Bedka et al. (2016) tried to minimize the use of fixed detection criteria. They developed two OT detection algorithms based on IR and VIS channels, and an OT probability was produced through a pattern-recognition scheme. The pattern that the scheme looks for is protrusion through the anvil caused by strong updrafts. Another pattern that is obvious in mature convective clouds with or without OT is "lumpy surface" from constant bubbling (Mecikalski and Bedka, 2006). Cloud top texture in VIS and IR channels has been explored using Spinning Enhanced Visible and Infrared Imager (SEVIRI) on Meteosat-8 satellite in Zinner et al. (2008) and Zinner et al. (2013), respectively. In addition to evaluating spatial texture, Müller et al. (2019) explores spatio-temporal gradients of water vapor channels in SEVIRI to estimate updraft strength. This study suggests a different way to calculate spatial gradients of visible channel in GOES-R to detect convection.

The use of VIS and IR sensors in detecting convection can benefit significantly with the launch of National Oceanic and Atmospheric Administration's (NOAA's) GOES-R Series which have high resolution, rapidly updating (i.e. 1 minute) imagery. This study makes use of this new data, namely the 1 minute data available from GOES-16 and GOES-17 in "mesoscale sectors" to update methods for detecting convection in different stages. Mesoscale sectors are manually moved around to observe interesting weather events. One is developed for CI using $T_b$ from an IR channel in GOES-R. As in previous papers measuring clout top cooling rate, temporal trends of the data were used, but since GOES-R has high temporal resolution, ten consecutive data with 1-minute interval were used. It has been challenging to correctly track convective clouds with 15 or 30-minute interval data which have been used in previous studies due to changing shape of convective clouds and merging or splitting of clouds. However, since clouds do not change as much within one minute, using one-minute data eliminates some of the errors from cloud movements that needed to be dealt with in some previous studies, and cooling rate is calculated applying linear regression on 1-minute data over 10 minutes, rather than using $T_b$ difference between 15 minutes. Another one is developed for mature convection using both reflectances from a VIS channel and $T_b$ from IR channels. For this algorithm, lumpy and rapidly changing surface and high cloud top height from mature convective clouds were used to detect clouds both with and without OTs. Lumpiness is calculated using Sobel operator which is an edge detection filter in image processing, and the lumpiness is explored at each minute throughout 10 minutes to look for regions with continuous bubbling. These two methods were then combined to provide detection of convection in all stages. The above methods are not intended to replace ground-based radars where these are available. Instead, the focus here is complementing ground-based networks, either off-shore or other regions lacking coverage.

The datasets that were used to detect convection and validate the results are described in Sect. 2, while the methods used to identify initial and established convection are explained in Sect. 3. Sect. 4 highlights the results of each method. Two case studies were examined followed by a one-month statistical study to quantify the operational accuracy of the methods.

## 2 Data

### 2.1 The Geostationary Operational Environmental Satellite R series (GOES-R)

Earth-pointing instruments of GOES-R consist of the Advanced Baseline Imager (ABI) with 16 channels, and the Geostationary Lightning Mapper (GOES-R Series Data Book, 2019). GOES-16 is the first of the two GOES-R series satellites to provide data for severe weather forecast over the United States and surrounding oceans (Schmit et al., 2017). Both $T_b$ and reflectance data from the ABI were used to detect convective regions. Mesoscale data with one minute temporal resolution were used to fully exploit its high temporal resolution of the new instrument.

Reflectance at 0.64μm (Channel 2) and $T_b$ at 6.2μm (Channel 8), 7.3μm (Channel 10), and 11.2μm (Channel 14) were used in the study. Channel 2 is a "red" band with the finest spatial resolution of 0.5km. This fine spatial resolution is useful to resolve lumpy, or bubbling surfaces of clouds in their mature stage. Channel 2 reflectance data were normalized by solar zenith angle so that a single threshold can be used throughout the method regardless of locations of the sun. Channel 14 is an IR longwave window band, which is a good indicator of the cloud top temperature for cumulonimbus clouds (Müller et al., 2018). High reflectance and texture of the cloud top seen in channel 2 and cloud top height inferred from channel 14 are combined to determine locations of mature convective clouds.

Channel 8 and 10 are ABI water vapor channels with 2km spatial resolution. Because Channel 8 sees WV at somewhat higher altitudes than Channel 10, they can observe WV associated with updrafts as clouds develop upwards, and were therefore used to detect early convection.

### 2.2 NEXRAD and MRMS

Multi-Radar/Multi-Sensor (MRMS) data developed at NOAA's National Severe Storms Laboratory were used for validation purposes. MRMS integrates the radar mosaic from the Next Generation Weather Radar (NEXRAD) with atmospheric environmental data, satellite data, lightning, and rain gauge observations to produce three dimensional fields of precipitation (Zhang et al., 2016). These quantitative precipitation estimation (QPE) products have a spatial resolution of 1km and temporal resolution of 2 minutes.

A "PrecipFlag" variable contained in the standard MRMS product classifies precipitating pixels into seven categories: 1) warm stratiform rain, 2) cool stratiform rain, 3) convective rain, 4) tropical–stratiform rain mix, 5) tropical–convective rain mix, 6) hail, and 7) snow. Details of the classification can be found in Zhang et al. (2016). It is a rather sophisticated classification of precipitation type as it not only uses reflectivity at various heights, but also takes into account vertically integrated liquid to distinguish convective core from stratiform clouds (Qi et al., 2013). A reduced set of these classes were used to validate the convective classification from GOES ABI data. In this study, warm stratiform rain, cool stratiform rain, and tropical-stratiform rain mix are all assigned a stratiform rain type while grid points with convective rain, tropical-convective rain mix, and hail are assigned a convective rain type. Along with the classification product, MRMS provides a variable called "Radar QPE quality index (RQI)". This product is associated with quality of the radar data, which is a combination of errors coming from beam

blockages and the beam spreading/ascending with range (Zhang et al., 2016). This flag is used to mask out regions with low radar

data quality. Only data with RQI greater than 0.5 are used in this study.

**3 Methodology**

This study examines methods to detect convective clouds at each life stage. Convective clouds can be divided into actively growing clouds and mature clouds. Actively growing clouds are usually clouds at the initial stage that grow nearly vertically while mature clouds are capped, but continue to bubble due to the release of latent heat. They often move horizontally after they

reach the tropopause. The proposed method to detect actively growing cloud is similar to previous CI studies mentioned in the introduction in the sense that the method uses temporal trends of $T_b$. The high temporal resolution data simplifies the method because the use of derived wind motion in tracking clouds is no longer necessary. One minute is short enough that cloud motion, at most, is to the adjacent grid points, and clouds can be easily tracked by focusing on overlapped scenes.

The method to detect mature convective clouds is similar to previous studies by Bedka et al. 2016 and Bedka et al. 2019 in terms of using the texture of the cloud top surfaces to infer strong updrafts. Cloud top surfaces of mature convective clouds are much bumpier than any other clouds, and their bumpiness is most evident in VIS images with the finest resolution. The following method uses horizontal gradient of reflectance to represent the bumpiness of cloud tops, and the magnitude of the gradients is used to distinguish convective cores from their anvil clouds. Cloud top temperatures from channel 14 are used to eliminate low

cumulus clouds that might appear bubbling.

**3.1 Detection of actively growing clouds with brightness temperature data**

In the early stage of convection, updrafts of water vapor eventually lead to condensation, the release of latent heat, and convective processes. Operational weather radars cannot observe small hydrometeors, but $T_b$ decrease at water vapor absorption bands of GOES-ABI is observed when these small hydrometeors start to develop. During the early convective stages, $T_b$s that are sensitive

to water vapor will decrease due to condensed cloud water droplets aloft generated by a strong updraft. Two ABI channels around the water vapor absorption bands, channel 8 (6.2μm) and channel 10 (7.3μm), were selected to cover water vapor updrafts at different height levels. These channels were used to find small regions consistent with developing clouds. If a cloud develops continuously for ten minutes and shows a large decrease in $T_b$ over ten minutes in either channel, the cloud is determined to be convective.


To compute the $T_b$ decrease in clouds, a window has to be defined as it is usually difficult to precisely define the boundary of clouds, especially at the early stages of convection. Since most of the early convective clouds are smaller than 10km in diameter, the window was defined as a 10km×10km box which is essentially a 5×5 matrix of satellite pixels consisting of 25 $T_b$s with 2km resolution. Considering the fact that a convective core usually has the lowest $T_b$ within its neighborhood, the $T_b$ matrix was

formed around a pixel only if that pixel had the lowest $T_b$ in the 5×5 matrix. However, this criterion alone could not distinguish convective cores from stratiform clouds and cloud edges which can also exhibit a local minimum. In addition to the lowest $T_b$, the shape of convective clouds is therefore also considered. As shown in the Fig. 1a, convective clouds not only have the lowest $T_b$ in their cores in all directions, but also have increasing $T_b$s away from the core, making their $T_b$ distributions look like an inverted two-dimensional (2D) Gaussian distribution. To select $T_b$ matrices that have this upside down Gaussian shape, an inverted 5×5

Gaussian matrix that has mean and standard deviation of the $T_b$ matrix was created and compared with the $T_b$ matrices. To focus the comparisons on the shape of the $T_b$ distribution (Fig. 1b), the maximum $T_b$ found in the 5×5 matrix was subtracted from all

values, and $T_b$ values were divided by the difference between maximum and minimum $T_b$ to normalize the $T_b$ matrix itself. If the $T_b$ matrix has a shape of a developing cloud (i.e. 2D upside down Gaussian), the absolute value of the difference between the $T_b$ matrix and the upside down Gaussian matrix will be small. A threshold of 10 for this absolute value of the difference between $T_b$

matrix and upside down Gaussian matrix (sum of residuals between normalized $T_b$ and upside down Gaussian) was empirically determined to exclude non-convective scenes. $T_b$ matrices with values greater than 10 are removed from the scene. This is done for all ten consecutive $T_b$ images that are one minute apart. Continuous overlaps of $T_b$ matrices for ten minutes imply that the cloud maintained a convective shape for ten minutes, and therefore, changes in $T_b$ are calculated to assess if the cloud in the $T_b$ matrices was growing. The minimum $T_b$s of the $T_b$ matrices at each time step were linearly regressed against time to measure a

decreasing trend. If the fitted line at each channel had a slope either smaller than -1K/min for channel 10 or -0.5K/min for channel 8, the grid point with the lowest $T_b$ at each time step for ten minutes as well as the neighboring 8 grid points in the window were classified as convective. This procedure is summarized in a flowchart in Fig. 2.

Water vapor channels have different sensitivity to water vapor, and thus, different values for the threshold are chosen for each

channel (channel 8 and 10). Since growth rate can vary depending on the surrounding environment and different evolution stages, it is important to find an appropriate threshold that best represents growth rate for clouds in their early stages. These thresholds are chosen based on the analysis of one-month data during July of 2017. The 5×5 $T_b$ windows that maintained the developing shape and had a decreasing trend of $T_b$ during ten minutes are collected over the one month period. A total of 38293 and 97042 (for channel 8 and 10, respectively) 5×5 windows that show decrease in $T_b$ were collected, and precipitation types from MRMS

were assigned for each window. Future MRMS convective flags up to 20 minutes after the detection period were included in the analysis because some time delays were observed in MRMS product when assigning convective flags, especially for early convection. When comparing GOES products to future MRMS products, future locations of GOES products were calculated assuming convection moves at the same speed that clouds moved during the initial ten minutes. Tables 1 and 2 show results applying different thresholds ranging from -0.1K/min to -2.0K/min. For each row, 5×5 pixel windows that show larger

temperature decrease than the corresponding threshold are collected, and they are analysed for potential convection. Numbers in the table represent the number of 5×5 windows that MRMS precipitation flags were assigned to either non-convective or convective at the corresponding 10-minute time window, as well as pixels that were flagged as convective by MRMS in the next 10 and 20 minutes to account for the fact that GOES can detect convection before the radar sees precipitation. However, not all the detection by the method is done early since MRMS can also sometimes assign early convection as convective before it

produces high reflectivity. The overall accuracy in the last column is calculated by dividing the number of windows that were convective within 20 minutes (sum of convective, convective within 10 min, and convective within 20 minutes) by the total number of the windows (sum of non-convective, convective, convective within 10 min, and convective within 20 minutes). Some convective clouds in the early stage show smaller decreasing trend than the thresholds, but using a smaller value for the threshold can introduce clouds that do not grow into deep convective clouds in the end. Clouds that develop into deep convective clouds are

eventually captured by these thresholds in later times as they show rapid intensification sooner or later. However, choosing a large cooling rate for the threshold will lead to less detection of convective clouds as not a lot of windows show large cooling rate. Therefore, thresholds of -0.5K/min and -1.0K/min for channel 8 and 10, respectively are chosen so that it detects reasonable amounts of convections. Cooling rate observed at channel 8 is smaller than channel 10 due to higher absorption at channel 8. Channel 8 senses moisture at higher altitude and thus, when water vapor starts to condensate at lower levels, it is less affected,

and its $T_b$ does not decrease as much as in channel 10. The matrix does not have to be detected at both channels, but using two

channels tends to find the same vertically growing clouds over time by detecting the cloud using channel 8 first and then using channel 10 later. This method will be called as growing cloud detection method hereinafter.

Furthermore, it is interesting to note that some clouds did not produce precipitation even with rapid growth over -2.0K/min (for channel 10). This would be due to mixing between convective cells and their dry environment or highly non-linear nature of chances of precipitation.

## 3.2 Detection of mature convective clouds with reflectance data

Mature convective clouds consist of convective cores and stratiform or cirrus regions where clouds have detrained from the core. The lack of discrete boundaries between different types of clouds makes it difficult to separate convective grid points from surrounding stratiform regions. Overshooting tops and enhanced-V pattern are well-known features in mature convective clouds, but these do not appear until their strongest stage and not in all convective clouds. Using such features associated with the deepest convective cores will create a detection gap between early and mature stages of convection. The method described here tries to minimize the gap, while still accurately detecting convective clouds.

Before evaluating the texture, only the grid points that are potentially parts of deep convection are selected using simple threshold values of VIS (ABI channel 2; 0.65μm) and IR (ABI channel 14; 11.2μm) channels. Channel 2 reflectance is highly correlated with the cloud optical depth (Minnis and Heck, 2012) while Channel 14 brightness temperature is related to cloud top temperature (Müller et al., 2018). These channels are used in GOES-R baseline product retrieval of cloud optical depth and cloud top properties, respectively. Any grid points with reflectance less than 0.8 or $T_b$ greater than 250K during ten time steps (10 minutes) are removed since they generally represent thin or low clouds such as cirrus or growing clouds that can be identified by the CI method described earlier. These thresholds are chosen rather generously to include some convective clouds that have not grown into deep convection yet, while still avoiding the misclassification of low cumulus clouds and thin anvil clouds as convective. The threshold of 250K is much warmer than typical values used in detecting deep convective features such as overshooting tops (Bedka et al., 2010) or enhanced-V (Brunner et al., 2007). Warmer threshold is intentionally chosen so that the method considers warmer convective clouds without those features in the next step when evaluating lumpiness of the cloud top. The choice of these thresholds is discussed in more detail in section 4.3.

Once cold, highly reflective scenes are identified, regions with bubbling cloud top are found. Bubbling cloud top is a distinct feature that appears in convective clouds, even in their early stages. The lumpiness of cloud tops can be numerically represented by calculating horizontal gradients in the reflectance field with the Sobel-Feldman (Sobel) operator which is commonly used in edge detection. The horizontal gradient is calculated at each pixel. The Sobel operator convolves the target pixel and its surrounding eight grid points with two kernels given in Eq. (1) to produce gradients in the horizontal and vertical direction.

$$G_x = \begin{bmatrix} +1 & 0 & -1 \\ +2 & 0 & -2 \\ +1 & 0 & -1 \end{bmatrix} \quad G_y = \begin{bmatrix} +1 & +2 & +1 \\ 0 & 0 & 0 \\ -1 & -2 & -1 \end{bmatrix} \qquad (1)$$

By using Eq. (2), gradients in each direction are combined to provide the absolute magnitude of the gradient at each point.

$$\text{Magnitude of gradient} = \sqrt{G_x^2 + G_y^2} \qquad (2)$$

Flat surfaces will have low gradients while cloud edges or lumpy surfaces will have high gradients. This lumpy feature is most evident in a VIS channel with the finest spatial resolution of 0.5km. IR fields are not very useful as the brightness temperature

variations in these lumpy surfaces tend to be quite small due to their relatively lower spatial resolution, and only cloud edges stand out.

The average of the horizontal gradients over the ten 1-minute time steps is calculated for each grid point, and grid points are removed if the average was less than 0.4 or greater than 0.9. Values below 0.4 or above 0.9 generally imply either stratiform

region with a flat surface or cloud edges with very high gradients, respectively. The thresholds are chosen to produce relatively low false alarms comparing results using other thresholds. Results using other thresholds are also shown in section 4.3 for a comparison. The remaining grid points were then interpolated into 1km maps to be consistent with the spatial resolution of MRMS dataset. Neighboring grid points were grouped to form clusters, and only the clusters with more than 5 grid points were assigned as a mature convective cloud to remove noise. This method will be called as mature cloud detection method hereinafter.

**4 Results and Discussion**

**4.1 June 28$^{th}$, 2017**

Supercell thunderstorms developed in Iowa and produced several tornado touchdowns. In Fig. 3a, deep convection had already developed over central Iowa at 19:30UTC, and two convective cells in the red box started to develop in southwest Iowa, although they do not stand out from surrounding low clouds in the VIS image. These two convective clouds became parts of major storm

system that formed around 21:30UTC, producing the tornadoes (Fig. 3b) in the area. MRMS Seamless Hybrid Scan Reflectivity (SHSR), which gives reflectivity at the lowest possible vertical level, is shown in Fig. 3c, and the MRMS PrecipFlag product is shown in Fig. 3d. Convection is colored in pink and stratiform in green. Although deep convections over central and northeast part of Iowa were assigned as convective in MRMS at 19:30UTC, the two growing clouds in the red box in Fig. 3a were not assigned convective flag until 19:48UTC.


Figure 4a shows brightness temperatures for ABI channel 10 (7.3μm) at 19:27UTC. The two growing convective cells in the blue box are shown in barely visible yellow surrounded by high $T_b$s. The one on the left was detected using 10-minute data from 19:25UTC, but since both clouds were detected together starting at 19:27UTC, a scene from 19:27UTC was used to demonstrate the method. Figure 4c and 4d show $T_b$ matrices that exhibited the correct shape for developing cells (Gaussian shape) at

19:27UTC and 19:36UTC. However, not all of the matrices in these figures showed the evolution of the developing cells (decreasing minimum $T_b$ over 10K) between the two time steps. The two matrices in the blue box satisfied both criteria of maintaining the shape of developing cells and growing vertically over ten time steps while other matrices did not satisfy either one of the criteria. These two matrices contain early convective clouds that grow into deep convection shown in Fig. 3b, and they are correctly captured by this method.


Results for the detection of mature convective clouds are shown in a step by step fashion in Fig. 5. Figure 5a is the same as in Fig. 3a, but is mapped using a different color table for better comparisons between steps. Figure 5b shows the pixels retained after eliminating all the grid points that did not meet the reflectance and $T_b$ thresholds (minimum reflectance over ten time steps greater than 0.8 and maximum $T_b$ over ten time steps less than 250K). Figure 5c shows the horizontal gradient values after

applying the Sobel operator. The colorbar is set to be within the range of 0.4 and 0.9 to display potential convective regions that passed these thresholds in colors. White regions are either regions that have average gradients greater than 0.9 such as cloud

edges or thin cirrus clouds, or regions that have average gradients less than 0.4 such as clear sky or stratiform regions. Eventually, only the regions that meet both the criteria in Fig. 5b and 5c are assigned to convection, and shown as white shade in Fig. 5d. Using reflectance threshold sometimes limits the detection of shaded convective regions that exhibits lower reflectance than the threshold of 0.8. This is case for the small imbedded white regions in the midst of high reflectance regions shown in Fig. 5b. However, these regions are relatively small, and once they are upsampled into 2km maps through nearest neighbour interpolation, some of these regions are included in the detection as shown in Fig. 5d.

Detection from GOES and MRMS is compared in MRMS's resolution of 1km, and in such high resolution, the location of a cloud seen from GOES and MRMS can be slightly different due to parallax displacement. For a better comparison between detection from GOES and MRMS, parallax correction based on Vicente et al. (2002) is applied to GOES detection using a constant cloud top height of 10km. Convective regions detected by GOES (Fig. 5d) are plotted with the parallax correction on top of the MRMS map (Fig. 3d), and it is shown in Fig. 6. When compared to high reflectivity regions in Fig. 3c and convective regions in Fig. 3d, convective regions, while not perfectly aligned due to a number of dynamic geometric reasons, do have a high degree of correspondence between the two detection methods. However, a straight line around 44N at the right edge of Fig. 5d is definitely not a convective region, and it is due to unrealistically high reflectance in the raw satellite dataset. These kinds of artifacts were removed later in section 4.3 when the method was applied to a full month of data. However, multiple lines are difficult to remove at this stage in the processing and will result in false alarm. As quality control procedures on ABI are improved, this may no longer be a source of significant errors.

## 4.2 June 18th, 2018

Another case was examined to evaluate the methods under different conditions. Severe storms developed over the Great Plains in June 18th, 2018, producing hail on the ground. At 22:30UTC, sporadic storms across Kansas and Oklahoma were observed by GOES-16. This scene contains both growing and mature convective clouds that are detected by MRMS during 22:30UTC - 22:40UTC period. Especially, four vertically growing clouds in this scene show different evolution and thus allow to elaborate more on the growing cloud detection method. MRMS PrecipFlag for the scene at 22:30UTC and 22:40UTC is shown in Fig. 7a and 7b, respectively. Green color represents stratiform and pink color represents convective clouds. Figure 7c and 7d are $T_b$ maps of the same scene at 22:30UTC and 22:40UTC, respectively. Growing clouds shown in purple, blue, yellow, and green boxes are detected by the growing cloud detection method, but all starting from different time. Times that each cloud is detected by GOES and MRMS are shown in Fig. 7a. Time for the growing cloud detection method is a period as the method uses 10 consecutive 1-minute data. Convection in the purple box was detected six minutes earlier than MRMS detection considering the last data used in the growing cloud detection method at 22:28UTC. Similarly, a cloud in the green box was detected a little earlier by GOES than MRMS. The growing cloud in the yellow box was detected at the same time by GOES and MRMS. On the other hand, the growing cloud in the blue box was detected later than MRMS detection at 22:38UTC. This cloud did not grow rapidly enough during 22:30UTC - 22:40UTC period as shown in $T_b$ maps of Fig. 7c and 7d and did not meet the $T_b$ threshold for channel 10 at the onset of convection. However, it was detected by channel 8 as it grew higher altitudes. This shows that a cloud that initially did not show high growth rate can have high growth rate as it vertically grows and can be detected by channel 8 later in time. These results show that even though the thresholds for the growing cloud detection method can miss some convective clouds that grow slowly in the beginning, the thresholds were adequate for detecting rapidly growing convective storms which are of more interest during the forecast.

Black regions superimposed on the brightness temperature map in Fig. 7c represent convective regions identified by the mature convection method, and Fig. 8 shows overlay figure of the black regions on top of the MRMS PrecipFlag map (Fig. 7a). There are slight misalignments of detected convective clouds between MRMS PrecipFlag products and GOES results possibly due to sheared vertical structures of the storms. One other thing to note here is that convective area detected by the mature cloud detection method is greater than what is detected in the previous case. This could be due to dependency of lumpiness on some geometrical considerations. Lumpiness is a function of the pixel spatial resolution, differences in optical depth and shadows. Spatial resolution decreases away from the equator, but higher solar zenith angles (due to altitude or time of day) not only increases optical depth, they also increase shadows. While this can of course be dealt with, it was ignored in this study which serves primarily as a proof of concept, as the method generally finds convective core correctly.



### 4.3 Statistical results with one-month data

Pixel-based validation of the two methods is conducted using one month of data during June of 2017. Results are validated against MRMS data as ground-based radar is used to detect convective regions during the short-term forecast, and precipitation is a rather direct indicator of convection in all stages. Since MRMS detection comprises convection in all stages, MRMS data are compared with GOES detection combining the two methods. Table 3 is a contingency table applying both methods to one month data and comparing in MRMS's grids with a spatial resolution of 1km. C represents convection detected by either GOES or MRMS, and NC represents non-convective regions. GOES-C/MRMS-C is "hits" that both MRMS and GOES methods detected as convective within 5km. In case of the growing cloud detection method, on the other hand, hits are defined if MRMS assigned convective within 30 minutes due to earlier detection by this method. GOES-NC/MRMS-C is "misses" that GOES missed detecting convection while MRMS assigned as convective. GOES-C/MRMS-NC is "false alarm" that GOES detected as convective, but MRMS did not. Lastly, GOES-NC/MRMS-NC is "correct negative case" that both MRMS and GOES did not detected as convective. From the contingency table, verification metrics of probability of detection (POD) and false alarm rate (FAR) can be calculated as below.



$$POD = \frac{hits}{hits + misses} \qquad FAR = \frac{false\ alarm}{hits + false\ alarm}$$


POD and FAR are useful tools in evaluating detection skill of a binary problem. POD and FAR calculated from Table 3 are 45.3% and 14.4%. Since POD and FAR can vary depending on the thresholds used in each method, choosing different thresholds is examined further.

Most of the detection are from the mature cloud detection method as mature convective clouds account for much larger area. The mature cloud detection method alone has FAR of 14.2% and POD of 43.7%. FAR and POD of the growing cloud detection method including 30-minute data are 22.2% and 3.9%, respectively. Relatively small FAR compared to FAR in Tables 1 and 2 (1- overall accuracy values) would be because Tables 1 and 2 are obtained based on each cloud while FAR and POD are calculated based on each grid point. Two PODs do not add up to 45.3%, POD from Table 3 due to overlapped detection. Since

the mature cloud detection method resort to several thresholds, results using different combinations of the three thresholds (reflectance at channel 2 and $T_b$ at channel 14 to remove shallow and low clouds, and horizontal gradients of reflectance at channel 2 to remove cloud edges as well as clouds with flat cloud top surfaces) are presented to show how they differ from the chosen thresholds. Two thresholds for cloud top texture, which are essentially horizontal gradients of reflectance, are evaluated

first. The upper threshold does not change results much (not shown), and cloud edges are effectively removed by the threshold of

0.9. The lower bound of the texture thresholds are varied, keeping the upper threshold and the $T_b$ and reflectance thresholds constant. Resulting FAR and POD are shown in Fig. 9. Using 0.5 (yellow) misses significant amounts of convective regions while using lower values (blue and red) substantially misclassifies stratiform regions with flat cloud tops as convective, although their PODs are much higher. Using 0.2 gives the closest results from the pixel-based validation in Zinner et al. 2013 using lightning data. However, FAR of 45.6% when using 0.2 is no different from a random chance of 50% that it is no longer useful,

while POD of 29.9% when using 0.5 will not give much information. Therefore, values of 0.4 and 0.9 (green diamond in Fig. 9) were chosen as a reasonable compromise between POD and FAR.

POD and FAR using different combinations of $T_b$ and reflectance thresholds are plotted in Fig. 10, and this time texture thresholds are kept constant with 0.4 and 0.9. The $T_b$ threshold is varied from 230K to 250K, and the reflectance threshold is

varied from 0.7 to 0.9. There is a trade-off between detecting more convective clouds that are transitioning into mature stage and incorrectly assigning cumulus clouds as convective clouds. Having lower value for the $T_b$ threshold or higher value for the reflectance threshold leads to small FAR, but also leads to small POD. To make this method effective and reduce FAR as much as possible for its potential use in the short-term forecast, 250K for the $T_b$ and 0.8 for the reflectance threshold (black diamond in Fig. 10a) are chosen. In data assimilation, it is preferable to provide no constraints than to provide the model with the incorrect

location of convection. A value of 240K and 0.7 (orange) also showed similar results, but 250K and 0.8 were chosen due to lower FAR.

Despite its FAR being relatively small, the method misses significant amounts of convective areas observed by MRMS. Therefore, regions that were missed are evaluated further to investigate which threshold contributed most to missing those

regions. Figure 11 shows histograms of $T_b$, reflectance, and texture in the convective regions that were missed by the above method. It is clear from the figure that the largest number of misses were due to low texture values (87.6% of all missed regions has lower gradients than 0.4). There are many reasons why convective regions appear to have flat cloud top surfaces. Anvil or thick cirrus clouds above convective regions can smooth out or cover bubbling cloud tops, and there is simply no way to avoid this problem. Another reason may be the nature of the classification method. Since classification by MRMS is determined by rain

rate, even if convective clouds are in a decaying mode and do not bubble anymore, clouds can still continue to precipitate considerable amounts, which would lead to convective category in the MRMS product. It is also possible that it is due to a misclassification of trailing stratiform regions using radars. It is indeed an ongoing research in the radar community since better convective/stratiform classification scheme improves QPE retrieval (Qi et al., 2013; Veljko et al., 2019).

As shown from these results, there are no perfect thresholds that can separate convective and stratiform clouds. Nevertheless, threshold values were chosen in line with our main objective - to avoid high FAR as much as possible and have decent POD comparable to radar products. Avoiding FAR is a higher priority than reaching higher POD as giving false information is most detrimental during data assimilation. Low FAR of 14.4% is achieved, and among those misclasified pixels, 96.4% of them are at least raining. Since the main objective of data assimilation is to have good initialization of precipitation, applying these methods

during data assimilation can still be beneficial in case the forecast model did not produce precipitation. Unfortunately, significant amounts of convective areas assigned by the radar product are missed. As shown in Fig. 11, most of the missed regions are excluded due to flat surface, and this is an intrinsic problem of using VIS and IR bands. If a convective cloud is developing in a less cloudy scene, it can be detected by the method most of the time. However, in case of a hurricane where cloud tops are rather

flat, or multi-layer clouds where cloud top information is decoupled from what is underneath, convection will be missed by the detection method. Furthermore, flat cloud top regions close to bubbling area might still be convective by MRMS due to high reflectivity, leading those regions to be classified as missed. The thresholds can be adjusted for other applications that may require higher POD.

## 5 Conclusion and summary

This study explores two methods to detect convective clouds in two different stages using GOES-R ABI data with one minute interval. Using such high temporal resolution data facilitates cloud tracking in a more effective way and helps reduce uncertainties coming from cloud tracking when calculating decreases in $T_b$ of the same cloud. Convective clouds in the early stage were detected using $T_b$s of ABI channels 8 and 10. These channels were used to find cloud scenes with the developing shape of convective clouds. They were then used again to calculate the $T_b$ decrease for those which maintained the developing shape for ten minutes. A cloud scene that had a consistent developing shape and a large decrease in $T_b$ over ten minutes was classified as convective by this method. Mature convective clouds were detected by masking out regions with high $T_b$ in ABI channel 14 and low reflectance in ABI channel 2 and finding regions with high horizontal gradients of reflectance over the course of ten minutes. Results from this mature cloud detection method were mostly consistent with the radar-derived products, although this method is limited to daytime use only. Nevertheless, it detects a wide range of convective area, not just regions with overshooting tops. Both methods are provided as a testing concept with several thresholds, and these thresholds can be tuned for an operational use if needed.

These methods work well for well-structured convective clouds, but there are limitations to this method as with most algorithms using IR and VIS sensors have. Cirrus cloud shields are the biggest problem as they block $T_b$ decreases underneath and smooth out lumpy reflectance surfaces. However, these methods can still be useful for defining convection for assimilation into models where radar data is not available. Because regions identified as convective are most likely convective (~85% accuracy (100%-FAR of 14.4%)), this can easily be assimilated while setting cloudy regions to "missing" since the accuracy of detecting convection under large cirrus shields is poor. Furthermore, results using Sobel operator, which is commonly used in image processing, implies that applying machine learning can be beneficial if the model can be set up to learn lumpy texture of convective clouds during training.

## Author contributions

All three authors designed the experiments. YL processed and analysed the data. CK and MZ gave feedbacks with their insights at every step of the data analysis. The manuscript was written jointly by YL, CK, and MZ.

## Competing interests

The authors declare that they have no conflicts of interests.

**Data availability**

NEXRAD reflectivity data were obtained by NOAA's National Centers for Environmental Information: doi:10.7289/V5W9574V. Past MRMS datasets are available at http://mtarchive.geol.iastate.edu/. GOES-R data were made available by Cooperative Institute for Research in the Atmosphere (CIRA).

**Acknowledgement**

This research is supported by CIRA's Graduate Student Support Program, as well as the WMO sponsored ICEPOP-18 project through the Korean Meteorological Administration.

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

(a)                              (b)

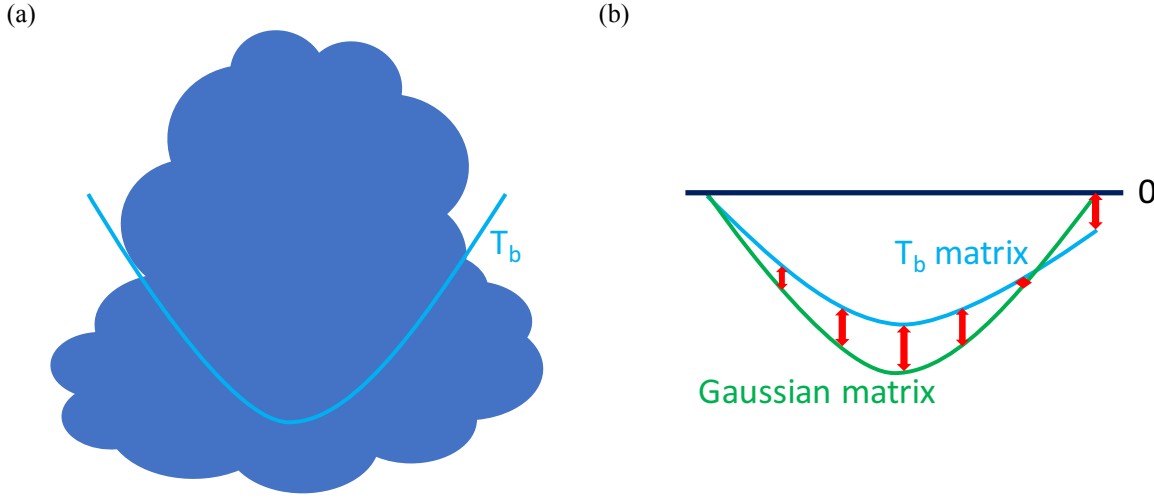

**Figure 1: (a) A typical shape of a convective cloud and its $T_b$ distribution around the convective core (blue line). (b) Schematic representation of distributions of the upside down Gaussian matrix (green) and the $T_b$ matrix (blue) when the cloud is convective.**




Grid points that show minimum $T_b$ in the 5×5 $T_b$ matrix are found.

Each 5×5 $T_b$ matrix is subtracted by the maximum $T_b$ value in the matrix and divided by the difference between maximum $T_b$ and minimum $T_b$.

Standard deviations of the 5×5 $T_b$ matrix in both directions are calculated and used to create upside down Gaussian matrix.

Calculate an absolute value of the difference between the $T_b$ matrix and the upside down Gaussian matrix for 10 time steps. If the values are smaller than 10 for consecutive 10 time steps, then decrease in minimum $T_b$ at channel 8 and 10 is calculated.

If the decreasing trend is either larger than -0.5K/min for channel 8 or −1K/min for channel 10, the middle and the neighboring 8 grid points are assigned as convective.

**Figure 2: A flowchart to summarize the growing cloud detection method.**



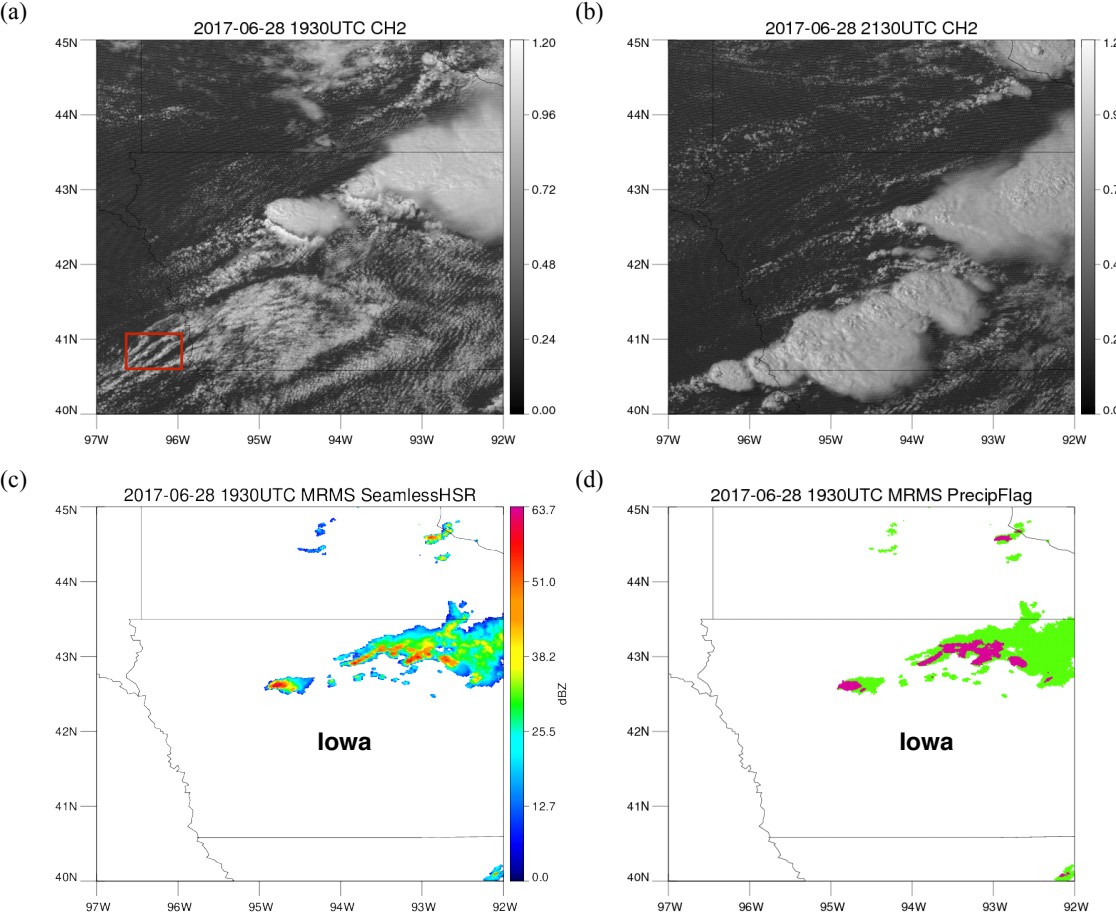

**Figure 3: (a) GOES-ABI 0.65µm visible channel imagery (0.5km) at 1930UTC 28 June 2017 over Iowa. Numbers on the**
**colorbar represent reflectances. The red box indicates regions where two convective cells are detected by the growing cloud**
**detection method. (b) GOES-ABI 0.65µm visible channel imagery at 2130UTC 28 June 2017. (c) MRMS Seamless Hybrid**
**Scan Reflectivity (SHSR) at 1930UTC 28 June 2017. (d) MRMS PrecipFlag at 1930UTC 28 June 2017. Pink represents**
**convective while green represents stratiform.**



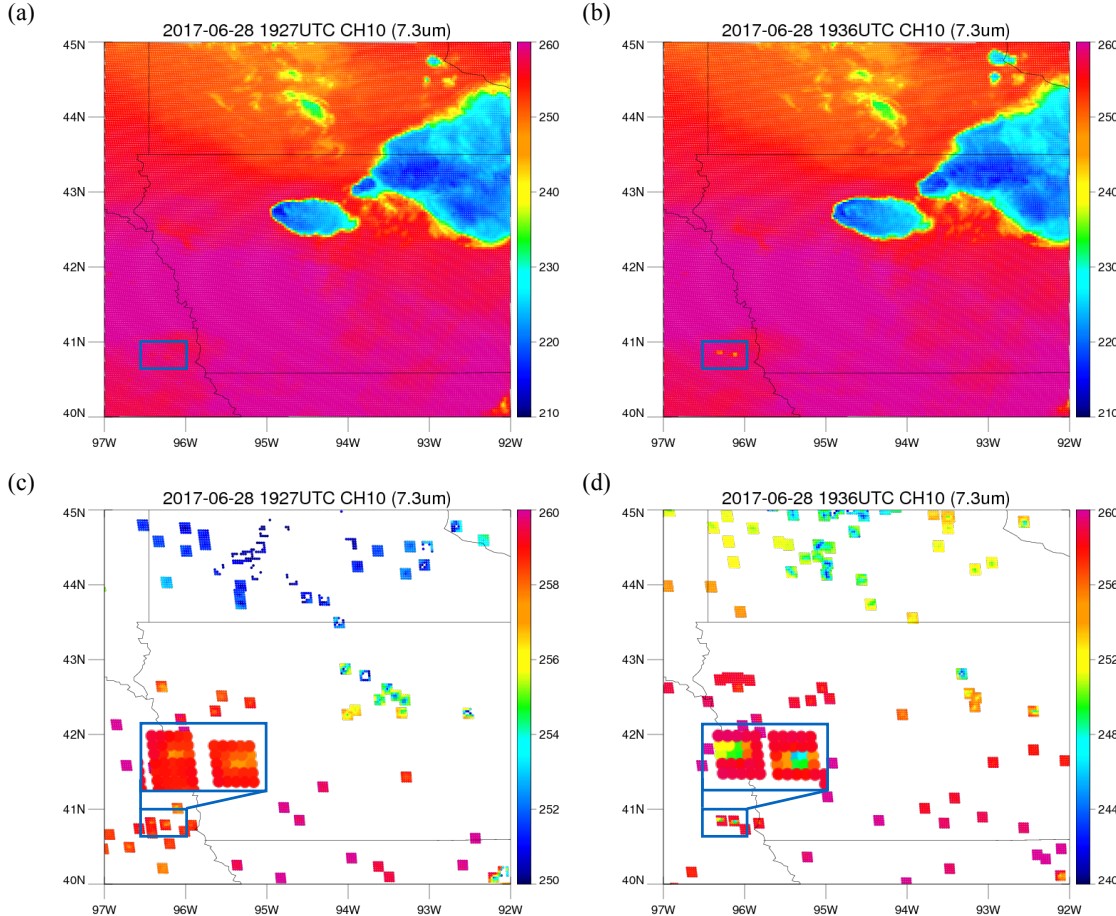

Figure 4: (a) GOES-ABI 7.3μm infrared channel imagery (K) at 1927UTC 28 June 2017. Blue box denotes regions where two convective clouds start to grow. (b) Same as Fig. 4a, but at 1936UTC. (c) $T_b$ matrices obtained from channel 10 (7.3μm) that have the Gaussian shape at 1927UTC 28 June 2017. Blue box denotes the same region as the blue box in Fig. 4a. Note that the scale of the colorbar is adjusted from Fig. 4a and 4b to better observe convective initiation. (d) Same as Fig. 4c, but at 1936UTC.



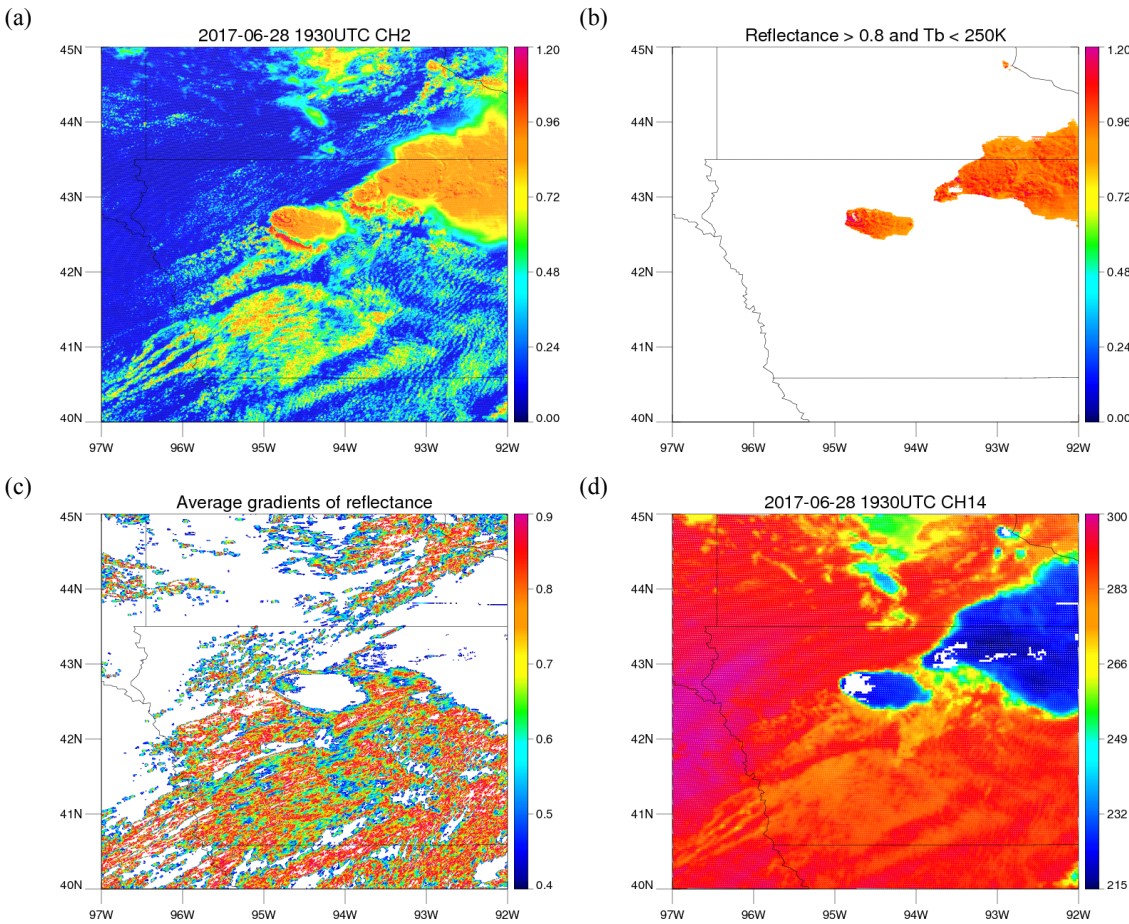


**Figure 5: (a) Same as Fig. 3a, but using different color table. (b) From the reflectance map in Fig 5a, regions that have reflectances over 10 minutes less than 0.8 or have $T_b$s greater than 250K over 10 minutes are assigned reflectance of zero, and therefore colored in white. (c) Map of average gradients of reflectances over 10 minutes. Regions with average gradient less**

**than 0.4 or greater than 0.9 are colored in white. (d) GOES-ABI 11.2μm infrared channel imagery (K) at 1930UTC 28 June 2017. Regions that passed two criteria from Fig. 5b and 5c are colored in white.**


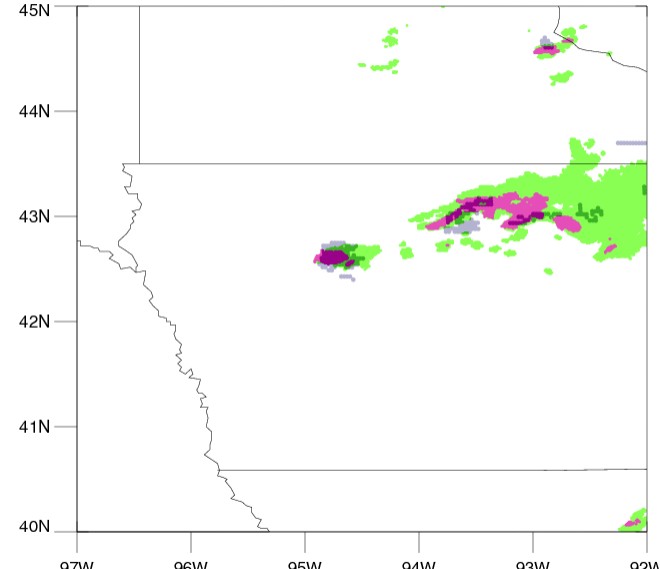

**Figure 6: Convective regions detected by GOES-16 (white regions in Fig. 5d) are colored in navy on top of MRMS PrecipFlag at 1930UTC 28 June 2017 (Same figure on Fig. 3d. Pink represents convective while green represents stratiform)**


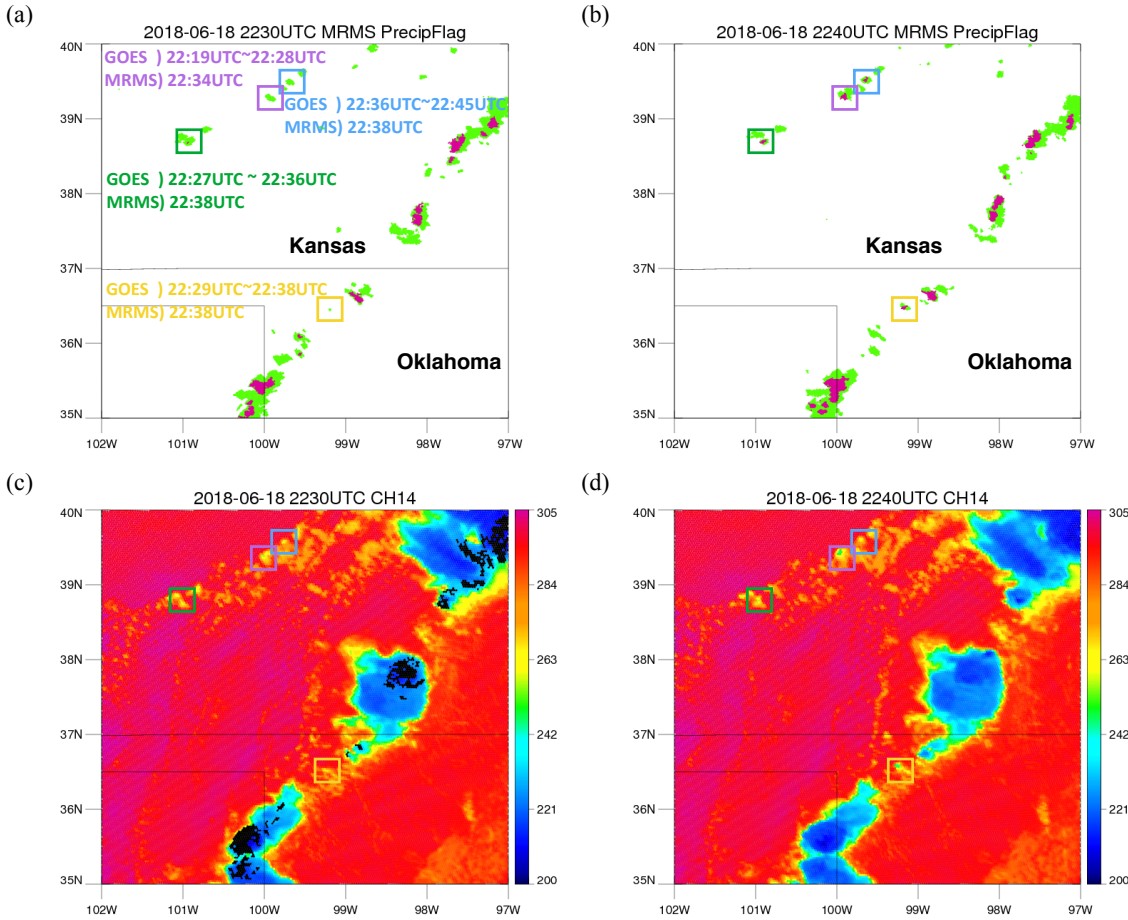

**Figure 7: (a) MRMS PrecipFlag at 2230UTC 18 June 2018. Pink represents convective while green represents stratiform. Times next to each box represent the times of GOES data used in the growing cloud detection method and time of detection by MRMS. (b) MRMS PrecipFlag at 2240UTC 18 June 2018. (c) GOES-ABI 11.2μm infrared channel imagery (K) at 2230UTC 18 June 2018 over the Great Plains. (d) Same as Fig. 7c, but at 2240UTC.**








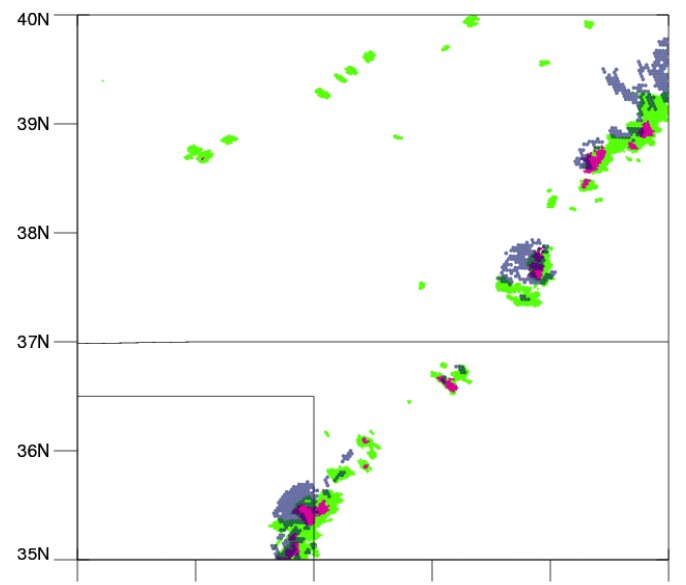

**Figure 8: Convective regions detected by GOES-16 (in Fig. 7c) are colored in navy on top of MRMS PrecipFlag at 2230UTC 18 June 2018 (Fig. 7a). Pink represents convective while green represents stratiform)**





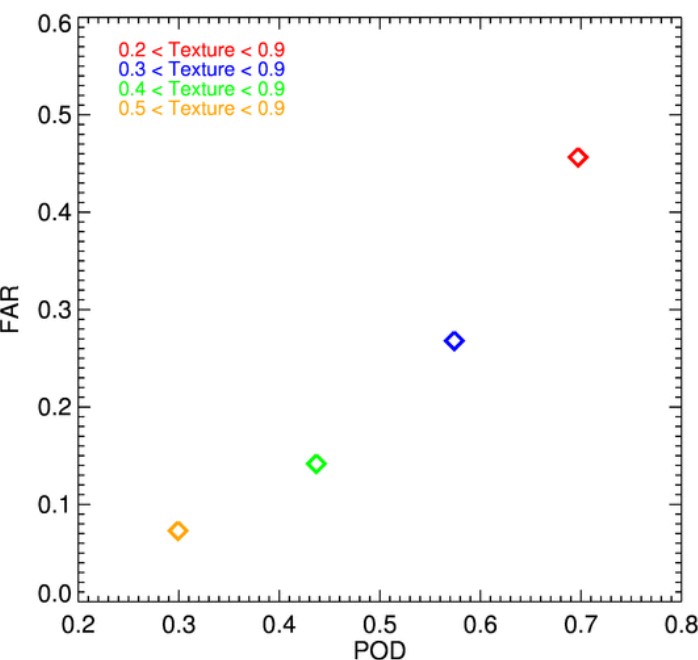

**Figure 9: Plot of probability of detection (POD) and false alarm ratio (FAR) using different texture thresholds of the mature cloud detection method. The $T_b$ and reflectance thresholds are kept constant with 250K and 0.8, respectively.**



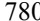

**Figure 10: Plot of probability of detection (POD) and false alarm ratio (FAR) for different combinations of T$_b$ and reflectance thresholds. The texture threshold of 0.4 and 0.9 are kept constant.**

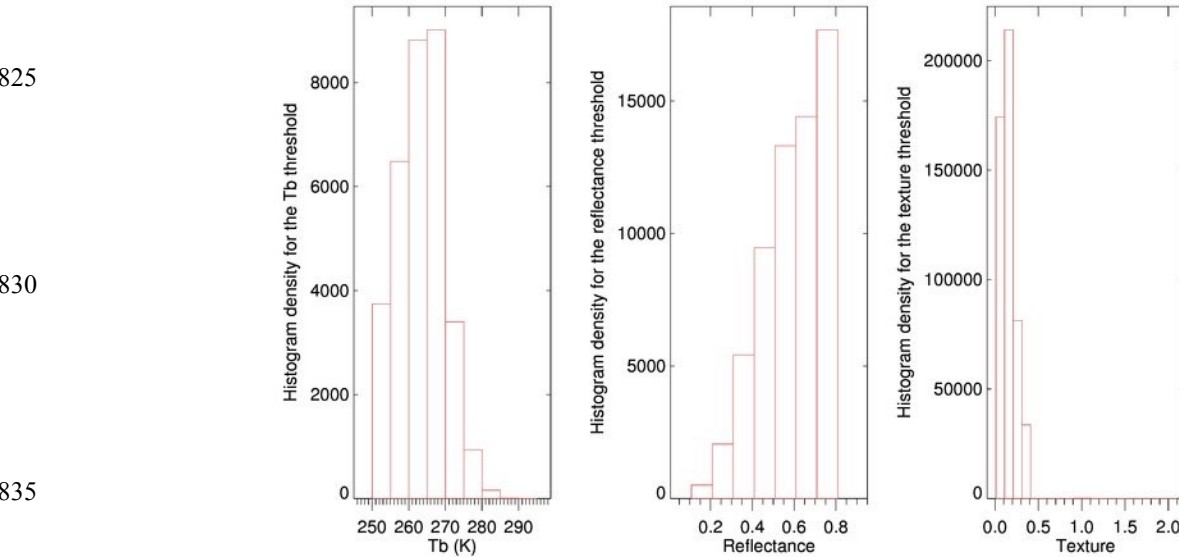

**Figure 11: Histograms of T_b, reflectance, and texture values if a pixel was assigned to be convective by MRMS, but not detected by the mature cloud detection method due to each of the thresholds.**

**Table 1. Number of non-convective, convective, convective within 10 minutes, and convective within 20 minutes for using different threshold values (channel 8)**

| Threshold value (K/min) | Non-convective | Convective | Convective within 10 min | Convective within 20 min | Overall accuracy |
|---|---|---|---|---|---|
| -0.1 | 3634 | 2911 | 250 | 89 | 47.2% |
| -0.2 | 740 | 2264 | 154 | 40 | 76.8% |
| -0.3 | 277 | 1831 | 117 | 28 | 87.7% |
| -0.4 | 153 | 1504 | 87 | 21 | 91.3% |
| -0.5 | 104 | 1266 | 87 | 16 | 92.8% |
| -0.6 | 67 | 1051 | 44 | 10 | 94.3% |
| -0.7 | 49 | 851 | 30 | 7 | 94.8% |
| -0.8 | 32 | 691 | 27 | 5 | 95.8% |
| -0.9 | 22 | 576 | 21 | 4 | 96.5% |
| -1.0 | 12 | 477 | 19 | 3 | 97.7% |
| -1.1 | 7 | 396 | 16 | 3 | 98.3% |
| -1.2 | 5 | 321 | 14 | 2 | 98.5% |
| -1.3 | 4 | 267 | 9 | 1 | 98.6% |
| -1.4 | 3 | 222 | 9 | 0 | 98.7% |
| -1.5 | 2 | 180 | 8 | 0 | 98.9% |
| -1.6 | 1 | 134 | 7 | 0 | 99.3% |
| -1.7 | 1 | 105 | 7 | 0 | 99.1% |
| -1.8 | 1 | 89 | 6 | 0 | 99.0% |
| -1.9 | 1 | 74 | 4 | 0 | 98.7% |
| -2.0 | 1 | 54 | 2 | 0 | 98.2% |

**Table 2. Number of non-convective, convective, convective within 10 minutes, and convective within 20 minutes for using different threshold values (channel 10)**

| Threshold value (K/min) | Non-convective | Convective | Convective within 10 min | Convective within 20 min | Overall accuracy |
|---|---|---|---|---|---|
| -0.1 | 21900 | 5041 | 1339 | 511 | 23.9% |
| -0.2 | 9225 | 3982 | 854 | 277 | 35.7% |
| -0.3 | 4357 | 3284 | 611 | 163 | 48.2% |
| -0.4 | 2241 | 2722 | 429 | 109 | 59.3% |
| -0.5 | 1234 | 2268 | 310 | 71 | 68.2% |
| -0.6 | 759 | 1954 | 233 | 40 | 74.6% |
| -0.7 | 479 | 1661 | 184 | 28 | 79.6% |
| -0.8 | 318 | 1430 | 139 | 22 | 83.3% |
| -0.9 | 22 | 576 | 21 | 4 | 86.1% |
| -1.0 | 147 | 1050 | 75 | 14 | 88.6% |
| -1.1 | 103 | 893 | 64 | 11 | 90.4% |
| -1.2 | 77 | 758 | 56 | 10 | 91.5% |
| -1.3 | 55 | 657 | 42 | 9 | 92.8% |
| -1.4 | 41 | 556 | 34 | 5 | 93.6% |
| -1.5 | 28 | 461 | 29 | 5 | 94.6% |
| -1.6 | 17 | 393 | 25 | 3 | 96.1% |
| -1.7 | 14 | 340 | 24 | 3 | 96.3% |
| -1.8 | 11 | 297 | 21 | 2 | 96.7% |
| -1.9 | 9 | 255 | 19 | 2 | 96.8% |
| -2.0 | 5 | 207 | 19 | 1 | 97.8% |

**Table 3. Contingency table of results applying both of GOES detection methods and validating against MRMS data during June of 2017. Pixel-based validation is conducted to produce this table. C and NC represent convective and non-convective, respectively. GOES-C/MRMS-C is "hits" that both MRMS and GOES methods detected as convective within 5km. GOES-NC/MRMS-C is "misses" that GOES missed detecting convection while MRMS assigned as convective. GOES-C/MRMS-NC is "false alarm" that GOES detected as convective, but MRMS did not. GOES-NC/MRMS-NC is "correct negative case" that**
**both MRMS and GOES did not detected as convective. Percentages in the parenthesis are obtained by dividing each number by the total number.**

|         | MRMS-C    | MRMS-NC    |
| ------- | --------- | ---------- |
| GOES-C  | 1759878   | 297291     |
|         | (2.73%)   | (0.46%)    |
| GOES-NC | 2125739   | 60244716   |
|         | (3.30%)   | (93.51%)   |
