# Peer review of "A simplified method for the detection of convection using high resolution imagery from GOES-16"

_Atmospheric Measurement Techniques, 2020_

## Referee Comment (RC1) · Anonymous Referee #2 · 24 Jul 2020

General comments:

There are some issues as below.

- No comparisons with past studies. It would also be feasible to compare with the results from ABI data of 15 min. on the same dates.

- It is hard to see the details in some figures.

- A table for accuracy is not found in the manuscript.

Specific comments:

Line 7: What is the meaning of the proper heating?

[Figure]

Line 9: Why is the latent heating especially mentioned here?

Line 11-12: Shouldn't it be more sensitive to the drop size?

Line 14: I don't understand how better spatial and temporal resolutions could be a solution to the intrinsic problem that optical sensors can only get information from the top layers of clouds.

Line 15: What are the life stages to be analyzed?

Line 17-18: Does this mean that the detection accuracy of the method for the clouds at early stages was 71%?

Line 19: How the rapid temporal evolution is identified? It needs to be clear, . . . rapid temporal evolution of what?

Line 21: Do the convective clouds here are clouds at all different life stages?

Line 22: It seems that the statement does not match with what is mentioned above in Line 14.

Line 26: What is 'this issue'?

Line 55: What does 'to initiate convection' mean?

Line 71: Is cooling really not seen in mature clouds? The sentence needs to be corrected.

Line 88: Where are the mesoscale sectors? What's the size?

Line 90-91: . . . ten consecutive data with 1-minute interval were used.

Line 89-94: It seems that the last part of the introduction is a bit detailed. They would be rather explained more concisely, and the details would be addressed in the method section.

Line 91: What are the errors from cloud movements?

[Figure]

Line 107: It seems that Table 1 is not really necessary. It can be removed and explained in the text.

Line 118-120: Need to add references. What about using channel differences? Past studies on detecting convective initiation have widely used channel differencing between water vapor channels and IR channels (c.f. Mecikalski2006, Lee2017)

Mecikalski, John R., and Kristopher M. Bedka. "Forecasting convective initiation by monitoring the evolution of moving cumulus in daytime GOES imagery." Monthly Weather Review 134.1 (2006): 49-78.

Lee, Sanggyun, et al. "Detection of deterministic and probabilistic convection initiation using Himawari-8 Advanced Himawari Imager data." (2017).

Line 140-141: Clouds do not necessarily reach the tropopause. Clouds form when air parcels reach the equilibrium level.

Line 142-143: How can the availability of higher temporal resolution data simplify the method to use two channels?

Line 159: What height does each channel usually reflect?

Line 189-190: "Clouds that develop into deep convective clouds are eventually captured by these thresholds in later times even if they had small decrease in the beginning." This sentence is a bit unclear.

Line 227-229: "It is intentionally chosen so that the method considers warmer convective clouds without those features in the next step when evaluating lumpiness of cloud top." This sentence is a bit unclear.

Results -> Results and Discussion

Line 253: It is almost impossible to see the Gaussian shape in Figure 3c. Maybe a close-up subfigure could be used here.

[Figure]

Line 258-259: "Since the same method is used in each time step, the same window can be captured throughout an overlapping time period despite the starting time. Therefore, this method can be used continuously in time." This sentence is a bit unclear.

Line 261-275: It would be better to move this paragraph to the beginning of this section.

Line 269-271: It would be much better to illustrate this as one Figure in the manuscript by merging both figures together.

Line 288-290: "These results show that even though the thresholds for the Tb method can be strict for some growing clouds, the thresholds were adequate for detecting convective storms in their earliest stages." This sentence is a bit unclear.

Line 299: "Since clouds do not grow at the same speed, . . .", which is a bit unclear.

Line 302: Why is the number of samples different?

Line 311: Why is it important to have the ability to detect convection earlier than radar? You mentioned earlier that the method of this study is to complement ground-based networks for either off-shore or other regions lacking coverage of radar data.

Line 318-322: It seems to be redundant.

Line 326: "The upper threshold does not change results much, . . ." The result for upper threshold is not shown here.

Line 328: However, the choice of 0.4 seems to lose a lot of convective regions.

Line 342-343: ". . . in preventing the method from detecting convective regions.", which needs to be corrected.

Figure 8 caption: ". . . due to only one of the thresholds.", which is a bit unclear.

Line 345: ". . . have flat cloud top surfaces." What percentage was this case? It would be good to provide quantitative values for one-month data.

Line 348: Q: Why are convective clouds in a decaying mode not considered?]

Line 349-350: "It is also possible that it is due to a misclassification of trailing stratiform regions using radars. Previous studies (Qi et al. 2013; Shusse et al. 2011) have indeed tried to improve the radar classification schemes." The sentences are a bit unclear.

Line 355-360: Reporting accuracy would be placed at the beginning of this section.

Line 354: to avoid FAR -> to avoid high FAR

Line 356: There is no Table 4 in the manuscript.

Line 360-362: However, further study on relating the detection of convection to precipitation is needed.

Technical corrections: - Tense in Abstract should be consistent.

- Line 15: ABI is not defined.

- Line 27: put a period after references.

- Line 42: . . . types using . . . -> . . . types by using . . .

- Line 67: Interest fields include . . . -> The interest fields include . . .

- Line 79: a feature of OT that it is . . . -> a feature of OT in that it is . . .

- Line 107: table 1 -> Table 1

- Line 148: Can't find Bedka et al. 2019 in References.

- Line 196: make -> makes

- Line 233: implies -> imply

---

## Referee Comment (RC2) · Anonymous Referee #1 · 28 Jul 2020

Review of "A simplified method for the detection of convection using high resolution imagery from GOES-16", by Yoonjin Lee et al.

The short manuscript tests some new and some not so new ideas for detection of thunderstorms in early and in mature stage in geostationary satellite data and compares it with ground-based radar results. A Gaussian cooling shape detection in water vapour GOES-16 imagery should provide the early stage detection, while a visible channel texture test should provide the mature stage detection. These ideas are not completely new, but in this manuscript they are applied to the new GOES-R data. Resolution of this data in time and space is much higher than that of many other geostationary satellite data. For this reason and because of some new aspects (Gaussian cooling detection) the manuscript is of interest for the community. My suggestion is to publish

the manuscript after major revision of the presentation.

The presentation has to be revised to make clear that not a full method nor a full verification is shown here, but just some experiments on details. Goals and limitations of the presented aspects have to be discussed in a more balanced way.

Major issues:

1. The literature overview is limited. Especially the cloud detection in satellite imagery over the last 20 years is widely ignored although ideas from this earlier work have found their way, at least indirectly, into this manuscript.

2. In some parts, the understanding of the underlying physics has to be discussed in more detail. For the first core concept, the information content of the WV channels 8 and 10, the discussion has to be improved at several places throughout the manuscript. Perhaps this method could even be further simplified by skipping the use of the less sensitive channel 8 data. For another central method, the visible channel texture, its limitations (at least for this manuscript's purpose) have to be evaluated and introduced in more detail.

3. Although the argument of convective precipitation information for data assimilation is touched upon in the beginning and shortly mentioned in the end again, the goal of the manuscript stays unclear. For quite some pages, the text seems to present a new, complete, very simple solution for a thunderstorm detection and early warning task. A task on which the satellite community has been working on for quite some 20 years. Major improvement seems to be reachable, because of the new GOES capabilities. Only when it comes to test cases and systematic verification, the limitations become obvious. This is where the simple solution presented would have to become more complex – as many working detection codes are. These limitations have to be discussed in the light of existing detection and nowcasting methods in literature as well as possible integration of the investigated aspects in such tools.
4. The "statistical results" section is not well presented up to now. Definitions and basis are not given clearly. Some of the statements and numbers seem questionable, in part, because of the limitations of the presentation.

Specifics and Minor:

l.20ff: In the light of the many questions left by the presentation of the statistical evaluation/tuning chapter, the numbers here are not useful. Either present some details of the used definitions and scoring basics or leave them out. I do not understand why there is one set of values for two independent methods (major 4).

l.28: Maybe you want to add something general like Gustafsson et al. 2018 or something very close to your motivation point like Scheck et al. 2020. (major 1)

l.84: The use of geostationary VIS and IR texture signals was introduced in automatic detection already by Zinner et al. 2008 (WV texture, Zinner et al. 2013). Another important tool forming an early reference for the use of IR and WV imagery and time trends in it is the EUMETSAT RDT algorithm (Morel and Senesi, 2002, Autones et al. 2009, Guillou et al 2011, see below). (major 1)

l.112ff: You state that Channel 2 data is "normalized by solar zenith angle". Please tell us how you do that. This is not a simple or straightforward task. You could normalize reflectivity, but for the texture signal cos(SZA) will not do the trick. The apparent lumpiness increases following a complex dependence on SZA and is strongly dependant on the cloud top structure. (major 2)

l.114f: Are you aware of Mueller et al 2019 "A Novel Approach for the Detection of Developing Thunderstorm Cells". That should be discussed somewhere. (major 1)

l.141: "GOES-R CI algorithm". Can you please give a reference?

l.145: Shouldn't "grids" be "grid cells". This sounds like lab slang to my non-native English ear.

l.155ff: "…updrafts of water vapour…", "…GOES-ABI … can." – You seem to formulate a misconception here. You cannot really see the rising water vapor. The signal is not strong enough. In a WV channel, you do see the water vapor background in mid-troposphere. You cannot see low-level dry-convection below condensation level. If you start to see convection cells in this data, it is the cloud body itself you see. Only once the cloud has formed, the emissivity is large enough to dominate the thermal signal in the WV channel. The cloud top "punches through" the background water vapor. Unless the mid-troposhere is very dry, you cannot see what's going on at lower. Please clarify and adjust the discussion here. (major 2)

l.176: "the difference between two matrices will be small.". Which two? Please clarify.

l.185: "smaller than -1K/min for channel 10 or -0.5K/min for channel 8". Why is there a difference? A growing cloud top is cooling at the same rate in both channels. Unless there still is considerable (colder) WV above it. Thus, it first shows up in the channel 10, later in the channel 8. You will increase the sensitivity of channel 8 to match channel 10 detections by lowering the slope threshold. You will earn a lot of uncertainty without adding any additional insight. Once the cloud top reaches the upper mid-troposphere above WV background, they will show exactly the same temperatures and trends. Please discuss, perhaps revise. (major 2)

l.221ff: Once more … What about low sun lumpiness? Shadows cast onto the cloud itself might dampen VIS reflectivity below 0.8. Please discuss. (major 2)

L.270: "… most of convective regions align well with high reflectivity regions in Fig. 2c…", You should not only talk about false alarms, but also about the POD. You are missing large regions with coldest temperatures and, thus, a quite obvious signal just next to the region you detected along 43 N and 93 W to 94 W! These regions shows up clearly in a cold absolute 11.2 mu data and in 11.2 lumpiness! This is opposed to your above statement on IR lumpiness and is a large area completely missed by your mature storm detection. Please discuss. (major 3)

[Figure]

l.281: "Growing clouds. . ." Are these boxes result of your method or did you place them by hand as marker to talk about certain areas. You are talking about the purple and blue boxes next. What about yellow and green? Did you miss them? Please make clear. (major 4)

l.295, Section 4.3, Statistical results: Please start this chapter with a clear definition of the "truth" you compare to, of a hit, miss, false alarm and false positive, all derived skill scores. What is the basic element of your scoring? Is it a grid point, a storm, or a 5x5 window? Please state that for all scores you derive for the early convection as well as the mature convection steps. Right now, this important information is (in part) hidden in the following chapter, but the reader has to guess most of the time. (major 4)

l.298ff: Again, it is still unclear for the reader, why you use both WV channels? Are there any channel 8 detection windows not contained in the channel 10 detected windows already? Please clarify or simplify.

l.303ff: "Future MRMS convective flags up to 30 minutes were included . . ." I do not fully understand. It was your goal to detect convection before the radar, wasn't it? That means, it is just logical to check the next 15 minutes/30 minutes. You should check the lliterature on MRMS and give us some details here. Using it, you have to discuss the choice of the future time span . . . the longer it is, the better your scores. (major 4)

l.306: Where do you get the "constant speed" from? Please add information.

l.310: What is the "accuracy" you are talking about? You have to introduce it. It seems to be the correct positives. Please clarify in the beginning of the chapter. (major 4)

l.311: "because most of early convection does not have such a strong updraft". No. It's because it is detected late. See my comment on the WV channels misconception above. In some situations, convection has to reach a considerable height before it can be detected. This is the reason why Mecikalski, Zinner or Guillou did not just use a WV channel to detect early stages. (major 2)

l.315f: The reasoning here is unclear. What about virga? I would just say, it is the typical turbulent, highly statistical nature of the chances of convective cells. Some just do not do it the moment latter.

l.335: For the reader, in order to be able to understand the impact on data assimilation, you have to give proper references or explain a lot more.

l.333f: "improvements in both FAR and POD (lower FAR and higher POD) when later data are included." This is not surprising and it is just tuning values. It would improve further, if you would include another 10 minutes, or even -10 minutes. Unless you can tell us a very good reason resulting from the function of the MRMS algorithm, I would suggest not showing the alternative numbers. They are not much different anyway. (major 4)

l.345ff: Checking of just one of the two examples you show, it is obvious how to improve it. In addition, the missed regions there are neither cirrus covered nor in decaying mode. You should accept and talk about shortcomings of your very simple method. There are good reasons out there that full detection and warning schemes are far more complex than your approach. Please discuss that. (major 3)

l.356: Where do these numbers come from? I cannot find them anywhere. They just show up here and in the abstract!?

l.366: Also, the next sentence is unclear. Are you talking about the early convection detection now? For the "early stage" detection, it makes sense to included "+30 min". For the "mature" detection, this would not be allowed.

Additional literature:

Y. Guillou, F. Autonès, S. Sénési, 2009, Detection and monitoring of Convective clouds by satellite, The Rapid Development Thunderstorm (RDT) product of the SAFNWC, WSN09, Whistler, 30 August–4 September 2009.

F. Autonès, J.-M. Moisselin, , 2010, Algorithm Theoretical Basis Document for "Rapid

Development Thunderstorms", Scientific documentation of SAF/NWC PGE 11 (RDT) v2011, code SAF/NWC/CDOP/MFT/SCI/ATBD/11, available on http://www.nwcsaf.org/

Morel C, Senesi S (2002) A climatology of mesoscale convective systems over Europe using satellite infrared imagery. I: Methodology. Q J Roy Meteorol Soc 128: 1953–71.

Müller, Richard & Haussler, Stéphane & Jerg, Matthias & Heizenreder, Dirk. (2019). A Novel Approach for the Detection of Developing Thunderstorm Cells. Remote Sensing. 11. 443. 10.3390/rs11040443.

Scheck, L., M. Weissmann, L. Bach, 2020, Assimilating visible satellite images for convective-scale numerical weather prediction: A case study, Q. J. R. Meteorol. Soc, https://doi.org/10.1002/qj.3840

Gustafsson, N., T. Janjic, C. Schraff, D. Leuenberger, M. Weissmann, H. Reich, P. Brousseau, T. Montmerle, E. Wattrelot, A. Bucanek, M. Mile, R. Hamdi, M. Lindskog, J. Barkmeijer, M. Dahlbom, B. Macpherson, S. Ballard, G. Inverarity, J. Carley, C. Alexander, D. Dowell, S. Liu, Y. Ikuta and T. Fujita, 2018, Survey of data assimilation methods for convective-scale numerical weather prediction at operational centres., Q. J. R. Meteorol. Soc. , 144, 1218-1256

Zinner, T., H. Mannstein, and A. Tafferner, 2008, Cb-TRAM: Tracking and monitoring severe convection from onset over rapid development to mature phase using multi-channel Meteosat-8 SEVIRI data, Meteorol. Atmos. Phys., DOI 10.1007/s00703-008-0290-y, 101, 191-210.

Zinner, T., C. Forster, E. de Coning, and H.-D. Betz, 2013, Validation of the METEOSAT storm detection and nowcasting system Cb-TRAM with lightning network data - Europe and South Africa, Atmospheric Measurement Techniques, 6, 1567–1583, doi:10.5194/amt-6-1567-2013.

---

## Author Comment (AC1) · 15 Aug 2020

We would like to thank two reviewers for their valuable comments and contributions to improve this manuscript.

*Table 1 (GOES-r channel description) will be removed based on reviewer 2's comment

*Table 4 that was not included in the manuscript by mistake will be added as Table 1 in the revised manuscript. (Table 1 is added in figure)

*Figure 3 is edited to have close-up subfigure.

*Figure 5 (laying GOES detection on top of MRMS detection) will be added based on both reviewers' comments

[Figure]

*Line number in the parenthesis is the number in the revised manuscript.

(-is Reviewer's comment and * is answer to that comment.)

General comments

-No comparisons with past studies. It would also be feasible to compare with the results from ABI data of 15 min. on the same dates. *The code that is used in previous studies is not publically available. Although previous studies also present their POD and FAR, they are validated against different dataset (lightning data or reflectivity threshold), and it makes it hard to compare with the previous studies.

- It is hard to see the details in some figures. *Resolutions of the figures will be updated.

- A table for accuracy is not found in the manuscript. *We apologize for that mistake. It will be added in the modified version.

Specific comments

-Line 7: What is the meaning of the proper heating? *'Proper heating' means reasonable heating to drive convection in the model. This phrase will be modified as 'The ability to detect convective regions and adding heating in these regions is the most important skill in forecasting severe weather systems.' (line 6).

-Line 9: Why is the latent heating especially mentioned here? *It was mentioned because in the operational model, after convection is detected, latent heating is added to drive convection, and methods developed in this paper are also intended to be used in the short-term forecast model. The purpose of this study is more clarified (line 29-36).

-Line 11-12: Shouldn't it be more sensitive to the drop size? *This sentence is changed to "Visible and Infrared sensors on a geostationary satellite can provide data that are more sensitive to small droplets" (line 11-12).

-Line 14: I don't understand how better spatial and temporal resolutions could be a solution to the intrinsic problem that optical sensors can only get information from the

top layers of clouds. *You're correct that it is not a perfect solution to the intrinsic problem. However, by having high spatial and temporal resolution, we can better observe clouds bubbling, which is an indicator of convective clouds. Even though we can't still see through inside of clouds, bubbling cloud top allows us to guess where convection is occurring, and it is the main feature used in this study to detect convective clouds.

-Line 15: What are the life stages to be analyzed? *Actively growing clouds in the vertical and mature convective clouds. This sentence will be modified to "This study develops two algorithms to detect vertically growing clouds and mature convective clouds using 1-minute GOES-16 ABI data." (line 15).

-Line 17-18: Does this mean that the detection accuracy of the method for the clouds at early stages was 71%? *Yes, but the accuracy increases when MRMS data up to 30min are included because MRMS tends to miss early convection with less or no reflectivity. However, these sentences in the abstract seem confusing and thus, will be modified based on changes made in 'statistical results' section.

-Line 19: How the rapid temporal evolution is identified? It needs to be clear, : : : rapid temporal evolution of what? *"the lumpy texture, and rapid temporal evolution" will be changed to "lumpy texture from rapid development" (line 19).

-Line 21: Do the convective clouds here are clouds at all different life stages? *'These convective clouds' meant convective clouds that are missed by ground-based radar. This sentence will be also changed based on changes made in 'statistical results' section.

-Line 22: It seems that the statement does not match with what is mentioned above in Line 14. *The authors are aware of the intrinsic problem of VIS and IR sensors as mentioned above. Thus, we wanted to address here that this intrinsic problem can't be solved.

-Line 26: What is 'this issue'? *This issue meansÂǎ'initiating convection in the right

location and intensity'.

-Line 55: What does 'to initiate convection' mean? *In the short-term forecast model, if radar echo exceeds certain values, latent heating is added to increase buoyancy in the atmosphere and trigger or initiate convection.

-Line 71: Is cooling really not seen in mature clouds? The sentence needs to be corrected. *The whole sentence in line 70 will be changed to "Convective clouds in their mature stage sometimes do not grow much in the vertical, and Tb decrease is not a main feature that is applicable to such clouds." (line 73).

-Line 88: Where are the mesoscale sectors? What's the size? *'Mesoscale sectors' are moved around manually by a person whenever there is an interesting weather event. Its size is 2000x2000 (1000kmx1000km) for channel 2 and 500x500 for IR channels.

-Line 89-94: It seems that the last part of the introduction is a bit detailed. They would be rather explained more concisely, and the details would be addressed in the method section. *It was briefly mentioned here to describe how the methods presented in this study differ from previous studies. Literature overview is revised (line 61-70 and line 84-89) and more references are added with help from reviewer1.

-Line 91: What are the errors from cloud movements? *Previous studies use atmospheric motion vector to track clouds, and there can be errors in its algorithm.

-Line 107: It seems that Table 1 is not really necessary. It can be removed and explained in the text. *As you suggested, this table will be removed because they're already explained in section 2.1.

-Line 118-120: Need to add references. What about using channel differences? Past studies on detecting convective initiation have widely used channel differencing between water vapor channels and IR channels (c.f. Mecikalski2006, Lee2017) *Mecikalski paper was already mentioned in the introduction, but since Lee 2017 paper wasn't mentioned, this paper will be added in the introduction. As mentioned above,

literature review will be revised.

-Line 140-141: Clouds do not necessarily reach the tropopause. Clouds form when air parcels reach the equilibrium level. *You're correct that not all clouds reach the tropopause. However, when clouds are mature enough, they sometimes reach tropopause and move horizontally, rather than vertically. (refer to Zinner et al., 2013; Validation of the Meteosat storm detection and nowcasting system Cb-TRAM with lightning network data – Europe and South Africa)

-Line 142-143: How can the availability of higher temporal resolution data simplify the method to use two channels? *It's worded in a wrong way. This sentence will be removed.

-Line 159: What height does each channel usually reflect? *There is no certain height that can represent each channel since height can vary depending on the situation. However, as mentioned in line 125∼127, the height that channel 8 represents is higher than channel 10 because channel 8 is more sensitive to water vapor. Figure 4 below (Figure_wv_height) is a "realtime" weighting function obtained from https://cimss.ssec.wisc.edu/goes/wf/. You can see from this figure that weighting function of channel 8 peaks at higher altitude.

-Line 189-190: "Clouds that develop into deep convective clouds are eventually captured by these thresholds in later times even if they had small decrease in the beginning." This sentence is a bit unclear. *This sentence will be changed to "Clouds that develop into deep convective clouds are eventually captured by these thresholds in later times as they show rapid intensification sooner or later." (line 196-197).

-Line 227-229: "It is intentionally chosen so that the method considers warmer convective clouds without those features in the next step when evaluating lumpiness of cloud top." This sentence is a bit unclear. *This sentence is changed to "Warmer threshold is intentionally chosen so that the method considers warmer convective clouds without those features in the next step when evaluating lumpiness of cloud top." (line 233-234).

-Results -> Results and Discussion *It is changed to Results and Discussion.

-Line 253: It is almost impossible to see the Gaussian shape in Figure 3c. Maybe a close-up subfigure could be used here. *Subfigure is added.

-Line 258-259: "Since the same method is used in each time step, the same window can be captured throughout an overlapping time period despite the starting time. Therefore, this method can be used continuously in time." This sentence is a bit unclear. *These sentences are removed.

-Line 261-275: It would be better to move this paragraph to the beginning of this section. *Paragraphs before these sentences were descriptions of the scene used to derive the results, and we think that they should come before the results.

-Line 269-271: It would be much better to illustrate this as one Figure in the manuscript by merging both figures together. *Figure 5 that shows convective regions by GOES on top of MRMS convective regions is added in the manuscript with the description of the figure (line 276-278).

-Line 288-290: "These results show that even though the thresholds for the Tb method can be strict for some growing clouds, the thresholds were adequate for detecting convective storms in their earliest stages." This sentence is a bit unclear. *It was changed to "These results show that even though the thresholds for the Tb method can miss some convective clouds that grow slowly in the beginning, the thresholds were adequate for detecting rapidly growing convective storms which are of more interest during the forecast." (line 296-298).

*Note that the whole section 4.3 will be modified based on two reviewers' comments. It will be modified to present one-month results first and then discuss results choosing different thresholds for the methods. Definitions of POD and FAR will be provided in the beginning of this section. The authors agree that providing results using different period of MRMS data in the validation separately can be confusing to readers. Therefore,

in the revised manuscript, only one set of POD and FAR will be presented from the contingency table that is recreated validating results from the reflectance method against 10 minute MRMS data and results from the Tb method against 30 minute (including future 20minute) MRMS data.

-Line 299: "Since clouds do not grow at the same speed, : : :", which is a bit unclear. *It is changed to "Since growth rate can vary depending on the surrounding environment and different evolution stages" (line 361).

-Line 302: Why is the number of samples different? *It is because channel 8 and 10 have different sensitivity to water vapor and represent different height of moisture.

-Line 311: Why is it important to have the ability to detect convection earlier than radar? You mentioned earlier that the method of this study is to complement ground-based networks for either off-shore or other regions lacking coverage of radar data. *Radar reflectivity is observed from bigger drops (eg. rain), and thus it takes time for radar to observe signals as cloud water becomes rain. On the other hand, Tbs are sensitive to water vapor, and they are expected to observe condensation by updrafts of water vapor before cloud water becomes rain. It is important to detect convection as early as possible because the ultimate goal of convection detection is to help forecast models to produce precipitation in the right place at the right time.

-Line 318-322: It seems to be redundant. *It will be put in the parenthesis (line 324-326).

-Line 326: "The upper threshold does not change results much, : : :" The result for upper threshold is not shown here. *We thought that it is not necessary to put figure for this because it did not have much change. But '(not shown)' will be added (line 330).

-Line 328: However, the choice of 0.4 seems to lose a lot of convective regions. *We preferred to have less POD and less FAR for its potential use in the short-term forecast model.

-Line 342-343: ": : : in preventing the method from detecting convective regions.",
which needs to be corrected *This sentence sounds misleading so it will be changed to
"Therefore, regions that were missed are evaluated further to investigate which thresh-
old contributed most to missing those regions." (line 348-349).

-Figure 8 caption: ": : : due to only one of the thresholds.", which is a bit unclear. *It will
be changed to "Histograms of Tb, reflectance, and texture values when the pixel was
not detected by the GOES detecting method due to each of the thresholds." Note that
Figure 8 will be Figure 9 in the revised manuscript.

-Line 345: ": : : have flat cloud top surfaces." What percentage was this case? It would
be good to provide quantitative values for one-month data. *It was visually shown in
figure 8 (now figure 9 in the modified version of the manuscript), but as you suggested,
the percentage will be presented (line 350).

-Line 348: Q: Why are convective clouds in a decaying mode not considered? *The
main purpose of application of this method is to add latent heating in active convective
regions and produce precipitation in the forecast model. If convective clouds are in a
decaying mode, precipitation will slow down, and therefore no need to add heating.

-Line 349-350: "It is also possible that it is due to a misclassification of trailing stratiform
regions using radars. Previous studies (Qi et al. 2013; Shusse et al. 2011) have indeed
tried to improve the radar classification schemes." The sentences are a bit unclear.
*It will be changed to "It is also possible that it is due to a misclassification of trailing
stratiform regions using radars. It is indeed an ongoing research in the radar community
since better convective/stratiform classification scheme improves QPE retrieval (Qi et
al., 2013; Veljko et al., 2019)." (line 355-357).

-Line 355-360: Reporting accuracy would be placed at the beginning of this section.
*As the whole section is revised, it will be placed in the beginning.

-Line 354: to avoid FAR -> to avoid high FAR *It will be changed.

-Line 356: There is no Table 4 in the manuscript. *It will be added as table 1 since "POD and FAR" are now placed in the beginning.

*All "technical corrections" has been made.

[Figure]

[Figure]

**Fig. 1.** Figure3

[Figure]

**Fig. 2.** Figure5

|           | MRMS-C | MRMS-NC |
|-----------|--------|---------|
| GOES-C    | 2.73%  | 0.46%   |
| GOES-NC   | 3.30%  | 93.51%  |

**Fig. 3.** Table1

[Figure]

Fig. 4. Figure_wv_height

---

## Author Comment (AC2) · 15 Aug 2020

We would like to thank two reviewers for their valuable comments and contributions to improve this manuscript.

*Table 1 (GOES-r channel description) will be removed based on reviewer 2's comment

*Table 4 that was not included in the manuscript by mistake will be added as Table 1 in the revised manuscript. (Table 1 is added in figure)

*Figure 3 is edited to have close-up subfigure.

*Figure 5 (laying GOES detection on top of MRMS detection) will be added based on both reviewers' comments

*Line number in the parenthesis is the number in the revised manuscript.

(-is Reviewer's comment and * is answer to that comment.)

-Major 1 *The introduction will be revised to elaborate more on literature overview, including suggested references from reviewer1.

-Major 2 *More discussion will be added regarding WV channels, and limitations of visible channel method will be more discussed in the text.

-Major 3 *First paragraph in the introduction will be revised to clarify the purpose of the manuscript.

-Major 4 *The whole section of 'statistical results' will be modified to present one-month results first and then discuss results choosing different thresholds for the methods. Definitions of POD and FAR will be provided in the beginning of this section. The authors agree that providing results using different period of MRMS data in the validation separately can be confusing to readers. Therefore, in the revised manuscript, only one set of POD and FAR will be presented from the contingency table that is recreated validating results from the reflectance method against 10 minute MRMS data and results from the Tb method against 30 minute (including future 20minute) MRMS data.

-l.20ff: In the light of the many questions left by the presentation of the statistical evaluation/tuning chapter, the numbers here are not useful. Either present some details of the used definitions and scoring basics or leave them out. I do not understand why there is one set of values for two independent methods (major 4). *As mentioned in Major 4, now only one set of POD and FAR is provided for better readability. Since two methods are used to detect convection in different stages and validation dataset includes convection in all stages, results from the two methods are combined.

-l.28: Maybe you want to add something general like Gustafsson et al. 2018 or something very close to your motivation point like Scheck et al. 2020. (major 1) *Both papers will be added (line 29).   -l.84: The use of geostationary VIS and IR texture signals

was introduced in automatic detection already by Zinner et al. 2008 (WV texture, Zinner et al. 2013). Another important tool forming an early reference for the use of IR and WV imagery and time trends in it is the EUMETSAT RDT algorithm (Morel and Senesi, 2002, Autones et al. 2009, Guillou et al 2011, see below). (major 1) *These references will be added in the introduction (line 65 and 87).

-l.112ff: You state that Channel 2 data is "normalized by solar zenith angle". Please tell us how you do that. This is not a simple or straightforward task. You could normalize reflectivity, but for the texture signal cos(SZA) will not do the trick. The apparent lumpiness increases following a complex dependence on SZA and is strongly dependant on the cloud top structure. (major 2) *Channel 2 data was divided by cos(SZA). A measure of lumpiness used in this study is horizontal gradients of the cloud top surface and thus, shadows (low reflectance) or even brighter surface enhanced by the effect of complexity on SZA actually helps distinguish convective regions from flat cloud top surfaces that are less affected by SZA.

-l.114f: Are you aware of Mueller et al 2019 "A Novel Approach for the Detection of Developing Thunderstorm Cells". That should be discussed somewhere. (major 1) *It will be added (line 88).

-l.141: "GOES-R CI algorithm". Can you please give a reference? *It is no longer operational products but, it is based on Mecikalski and Bedka (2006) or Mecikalski et al. (2010) cited in this paper.

-l.145: Shouldn't "grids" be "grid cells". This sounds like lab slang to my non-native English ear. *The term will be changed to "grid points".

-l.155ff: ": : :updrafts of water vapour: : :", ": : :GOES-ABI : : : can." – You seem to formulate a misconception here. You cannot really see the rising water vapor. The signal is not strong enough. In a WV channel, you do see the water vapor background in midtroposphere. You cannot see low-level dry-convection below condensation level. If you start to see convection cells in this data, it is the cloud body itself you see. Only

once the cloud has formed, the emissivity is large enough to dominate the thermal signal in the WV channel. The cloud top "punches through" the background water vapor. Unless the mid-troposhere is very dry, you cannot see what's going on at lower. Please clarify and adjust the discussion here. (major 2) *We agree that these sentences were misleading. "It will be modified to Operational weather radars cannot observe small cloud water, but water vapor absorption bands in GOES-ABI, are more sensitive to these small droplets. During the early convective stages, Tbs that are sensitive to water vapor will decrease due to condensed cloud water droplets aloft generated by a strong updraft." (line 163-165).

-l.176: "the difference between two matrices will be small.". Which two? Please clarify. *It will be clarified to "the absolute value of the difference between the Tb matrix and the inverse Gaussian matrix" (line 182-183).

-l.185: "smaller than -1K/min for channel 10 or -0.5K/min for channel 8". Why is there a difference? A growing cloud top is cooling at the same rate in both channels. Unless there still is considerable (colder) WV above it. Thus, it first shows up in the channel 10, later in the channel 8. You will increase the sensitivity of channel 8 to match channel 10 detections by lowering the slope threshold. You will earn a lot of uncertainty without adding any additional insight. Once the cloud top reaches the upper mid-troposphere above WV background, they will show exactly the same temperatures and trends. Please discuss, perhaps revise. (major 2) *As you pointed out, the cooling rates are similar once clouds are mature enough because their Tbs themselves are similar. However, when clouds are in the very beginning stage, the Tb difference between channel 8 and 10 is high, and the cooling rate is observed to be different. This makes sense because in order to exhibit the same Tb at both channels at their mature stage, Tb at channel 10 which is usually lower than Tb at channel 8 has to increase faster than Tb at channel 8. Sentences "Growth rate observed at channel 8 is smaller than channel 10 due to higher absorption at channel 8. Channel 8 senses moisture at higher altitude and thus, when water vapor starts to condensate at lower levels, it is

less affected, and its Tb does not decrease as much as in channel 10. As clouds grow thicker, signals in water vapor absorption bands are dominated by the clouds, less from water vapor, and their Tbs becomes similar. Therefore, it makes sense again that the growth rate at channel 10 has to be bigger to catch up lower Tb in channel 8." will be added (line 376-380).

-l.221ff: Once more : : : What about low sun lumpiness? Shadows cast onto the cloud itself might dampen VIS reflectivity below 0.8. Please discuss. (major 2) *It will be discussed using Figs. 4b and 4d. Sentences "Using reflectance threshold sometimes limits detecting shaded convective regions that exhibits lower reflectance than the threshold of 0.8, and white regions surrounded by colored regions in Fig. 4b are such regions. However, these regions are relatively small, and once they are upsampled into 2km map with nearest neighbour interpolation, some of these regions are included in the detection as shown in Fig. 4d." will be added (line 271-274).   -L.270: ": : : most of convective regions align well with high reflectivity regions in Fig. 2c: : :", You should not only talk about false alarms, but also about the POD. You are missing large regions with coldest temperatures and, thus, a quite obvious signal just next to the region you detected along 43 N and 93 W to 94 W! These regions shows up clearly in a cold absolute 11.2 mu data and in 11.2 lumpiness! This is opposed to your above statement on IR lumpiness and is a large area completely missed by your mature storm detection. Please discuss. (major 3) *It seems like it wasn't clear from two figures being separate and thus, overlaying figure will be added as Figure 5 (with the description of the figure "For a better comparison between detection from GOES and MRMS, convective regions detected by GOES (Fig. 4d) are parallax corrected with a constant cloud top height of 10km and plotted on top of the MRMS map (Fig. 2d), and it is shown in Fig. 5. Most of convective regions align well with high reflectivity regions in Fig. 2c and convective regions in Fig. 2d." in line 276-278). You will see that over 43N and 93W to 94W, convective regions are also detected by the method.

-l.281: "Growing clouds: : :" Are these boxes result of your method or did you place

them by hand as marker to talk about certain areas. You are talking about the purple and blue boxes next. What about yellow and green? Did you miss them? Please make clear. (major 4) *Boxes are clouds that were detected by the Tb method. This sentence will be changed to "Growing clouds shown in purple, blue, yellow, and green boxes are detected by the Tb method, but all starting from different time." to make it clear for readers (line 288-289). Clouds in yellow and green boxes are discussed few sentences after this sentence (line 294 and 295)

-l.295, Section 4.3, Statistical results: Please start this chapter with a clear definition of the "truth" you compare to, of a hit, miss, false alarm and false positive, all derived skill scores. What is the basic element of your scoring? Is it a grid point, a storm, or a 5x5 window? Please state that for all scores you derive for the early convection as well as the mature convection steps. Right now, this important information is (in part) hidden in the following chapter, but the reader has to guess most of the time. (major 4) *The definition of POD and FAR and its application in this study will be added in the beginning of section 4.3 (line 307-315).

-l.298ff: Again, it is still unclear for the reader, why you use both WV channels? Are there any channel 8 detection windows not contained in the channel 10 detected windows already? Please clarify or simplify. *Yes, there were windows detected by one channel but not by the other channel. It will be discussed in the text using convective cloud in the blue box in the second case study. (line 293)

-l.303ff: "Future MRMS convective flags up to 30 minutes were included : : :" I do not fully understand. It was your goal to detect convection before the radar, wasn't it? That means, it is just logical to check the next 15 minutes/30 minutes. You should check the lliterature on MRMS and give us some details here. Using it, you have to discuss the choice of the future time span : : : the longer it is, the better your scores. (major 4) *As you pointed out, clouds detected by the Tb method are often detected earlier than radar while clouds detected by the reflectance method are usually detected at the detection time by radar, although figure 8 suggests that it still has an ability to detect

earlier. This was why two sets of POD and FAR were given in this section, and it was confusing for readers. Therefore, as mentioned earlier when major 4 was discussed, it will be changed to show one set of POD and FAR from combined results by validating results from the reflectance method using only 10 minute data and results from the Tb method using 30 minute (including additional future 20 minute) data. This result will be presented in the beginning of section 4.3.

-l.306: Where do you get the "constant speed" from? Please add information. *This sentence will be changed to "assuming convection moves at the same speed that clouds moved during the initial ten minutes" (line 368).

-l.310: What is the "accuracy" you are talking about? You have to introduce it. It seems to be the correct positives. Please clarify in the beginning of the chapter. (major 4) *It will be clarified "100% accuracy of detecting convection as in MRMS" (line 372).

-l.311: "because most of early convection does not have such a strong updraft". No. It's because it is detected late. See my comment on the WV channels misconception above. In some situations, convection has to reach a considerable height before it can be detected. This is the reason why Mecikalski, Zinner or Guillou did not just use a WV channel to detect early stages. (major 2) *This sentence was misleading as well. It will be changed to "it misses much of the convection and loses an ability to detect convection earlier than radar because not all convective clouds have such a strong updraft." (line 374).

-l.315f: The reasoning here is unclear. What about virga? I would just say, it is the typical turbulent, highly statistical nature of the chances of convective cells. Some just do not do it the moment latter. *It is modified to "This would be due to mixing between convective cells and their dry environment or highly non-linear nature of chances of precipitation." (line 381-382).

-l.335: For the reader, in order to be able to understand the impact on data assimilation, you have to give proper references or explain a lot more. *It is modified to "To make

this method effective and reduce FAR as much as possible for its potential use in the short-term forecast" line 339-340, and suggested reference (Gustafsson et al. 2018) is added in the introduction.

-l.333f: "improvements in both FAR and POD (lower FAR and higher POD) when later data are included." This is not surprising and it is just tuning values. It would improve further, if you would include another 10 minutes, or even -10 minutes. Unless you can tell us a very good reason resulting from the function of the MRMS algorithm, I would suggest not showing the alternative numbers. They are not much different anyway. (major 4) *The whole section of 'statistical results' will be modified to reflect this comment.

-l.345ff: Checking of just one of the two examples you show, it is obvious how to improve it. In addition, the missed regions there are neither cirrus covered nor in decaying mode. You should accept and talk about shortcomings of your very simple method. There are good reasons out there that full detection and warning schemes are far more complex than your approach. Please discuss that. (major 3) *Its limitations are elaborated in section 4.3 using different thresholds. As mentioned in section 4.3, most of the missed regions had a flat cloud top surface, which is a key feature of stratiform clouds in this study. And since the results are compared with radar products observed from the ground, convectively raining pixels might not perfectly align with bubbling pixels. However, as you can see from the two case studies, the locations of convective cloud clusters detected by GOES and MRMS are very close to each other, and most convective regions in each scene are detected by GOES. Nevertheless, limitation regarding the reflectance threshold will be added based on your comment in "l.221ff" and it will be mentioned again in the last paragraph of section 4.3 (line 384-388).

[Figure]

**Fig. 1.** Figure3

[Figure]

**Fig. 2.** Figure5

|          | MRMS-C | MRMS-NC |
|----------|--------|---------|
| GOES-C   | 2.73%  | 0.46%   |
| GOES-NC  | 3.30%  | 93.51%  |

**Fig. 3.** Table1

---

## Referee Report (RR1)

I enjoyed reading "A simplified method for the detection of convection using high resolution imagery from GOES-16". The authors did a good job responding to my previous comments. But I think this paper needs some revision before being accepted, as I will describe in detail below.

Line 7: As a suggestion, "The ability to detect convective regions and to add latent heating to drive convection in weather forecast models is the most important skill in forecasting severe weather systems."

Line 12-15: As a suggestion, "Relatively new geostationary satellites, Geostationary Operational Environmental Satellites-16 and -17 (GOES-16 and GOES-17), along with Himawari-8, can make up for this lack of vertical information through the use of very high spatial and temporal resolutions, allowing to better observe bubbling features on convective cloud tops."

Line 20: Please provide a long name of MRMS.

Line 21: Please specify accuracy measures reported here.

Line 46: The 'peakedness' and 'surrounding area' criteria are not well explained by the following sentence.

Line 50: Please replace "at -10°C or higher" to "at -10°C height or above"

Line 50-51: Does Zhang and Qi (2010)'s method use the threshold for convective precipitation?

Line 62: Tb has not yet been defined.

Line 87: Please specify which Meteosat series.

Line 93: Please explain how "mesoscale sectors" is defined.

Line 96 "errors from cloud movements": Could you please elaborate on the errors from cloud movements?

Line 99: Change "Tb" to "Tb from IR channels"

Line 140-142 "It is a rather sophisticated classification…": Shouldn't this sentence be moved after the sentence "Details of the classification can be found in Zhang et al. (2016)." in line 137.

Line 182 "inverse Gaussian": It might be misleading with the term, inverse Gaussian distribution used in probability theory. Please use a different term.

Line 183: Please change "Tb shape" to "the Tb matrix"

Line 190-191 "Since one-minute data can be noisy, the decreasing trend was considered instead of an actual difference in Tb during 10minutes.": would leave this sentence out.

Line 199 "Using two channels help find the same clouds in different levels.": How are both channels used to find the same clouds? Does it mean that both channels need to satisfy the conditions? Please clarify, and change "help" to "helps".

Line 202: Change "make" to "makes"

Line 221: Change "relative to" to "due to"

Line 223: Please correct "texturs".

Line 236: It would be better to put the paragraph for the Sobel operator after the paragraph that describes screening scenes with VIS and IR channels (i.e., lines 223-234).

Line 238: Please mention how the thresholds, 0.4 and 0.9, were obtained. Change "implies" to "imply"

Line 239: "... with very high gradients." to "... with very high gradients, respectively."

3. Methodology: It would be good to have a flowchart for the methodology so that readers can have an overview of the methods used in this study.

Line 246: Isn't it southwest, not southeast?

Line 248-249: The radar data could be described in section 2.2, and the sentence would be something like "The two cells appeared in the composited NEXRAD radar data…"

Line 254: "channels 10" to "channels 10 (7.3 µm)". Please correct "the white circle".

Line 259-260: Change "The only two matrices in this scene that satisfied both criteria of maintaining the shape of developing cells and growing vertically over ten time steps were the two in blue circles." to "The two matrices in the blue box satisfied both criteria of maintaining the shape of developing cells and growing vertically over ten time steps.", or please revise it better.

Line 276: Please describe what parallax correction is and how it is calculated/corrected.

Line 289 "from different time": It would be better to provide the times when each coloured box was detected.

Line 291: "minute" to "minutes"

Line 293 "This shows a need to use both channels in the detection.": Does it mean that both channels need to be satisfied together in the detection? Please clarify.

Line 296 "the Tb method": It would be good to name the two methods for growing cloud detection and mature cloud detection method, respectively, in the Methodology section beforehand and use them in the following discussions.

Line 300 "Black regions superimposed on the brightness temperature map in Fig. 6c represent…": It would be good to also superimpose the detected mature convective clouds on the MRMS map, same as in Figure 5 for a better comparison.

4.3 Statistical results with one-month data: The part that explains the method seems to be quite huge, so it might be better to move them to the Methodology section.

Line 329: Using a threshold of 0.5 looks a reasonable compromise as well. It would be good to discuss the optimal balance between POD and FAR. It would be good to explain which one is the more important factor and why.

Line 333 "... mature convective clouds in the earlier stage…": Does it mean detecting clouds that grow into mature convective clouds?

Line 335-336 "... for its potential use in the short-term forecast, …": Is it okay to have small POD in the short-term forecast. Please give an explanation on it.

Figure 7: Please remove "(a)".

Line 337-338 "Figure 8b shows results including MRMS data 10 minutes after the detection period.": What does it mean to include MRMS data 10 min. after the detection period? Why were the data 10 min. after the detection period included? Please clarify.

Line 339-340 "Fig. 8b still shows its ability to detect convection earlier than MRMS.": It is unclear how Fig. 8b shows the ability of detecting convection earlier than MRMS.

Figure 9: Please add labels for x-axis with units.

Line 360: When all the windows were first collected, what were the thresholds used for each channel?

Line 365: Please add "... for channel 8 and 10, respectively." at the end.

Table 2 and 3: How are the first two columns obtained? How is the overall accuracy calculated? Please more clarify how the experiment is conducted for Table 2 and 3.

Line 375-376 "Therefore, it makes sense again that the growth rate at channel 10 has to be bigger to catch up lower Tb in channel 8.": Why does the growth rate at channel 8 needs to be caught by channel 10?

Line 377-378: Please give a reference.

Line 382-383: Could you explain more why false alarms are most detrimental to data assimilation? and please give a reference.

Line 385-387: What if the POD is so small that only a few convective clouds with most precipitation are detected? Could you please explain more on the last two sentences?

Line 390: "...to detect convective clouds" to "...to detect convective clouds in two different stages…"

Line 403 "extremely": It sounds a bit too strong. Please remove it or replace it with another word.

Results and Discussion: The labels in most figures are so small that can hardly be identified.

Please check typos and tense throughout the manuscript.

---

## Referee Report (RR2)

Review of version 2 of "A simplified method for the detection of convection using high resolution imagery from GOES-16", by Yoonjin Lee et al.

The manuscript is still not fit for publication. The presentation has been improved slightly, but still major issues have not been solved. Still I think, this could become an interesting paper after revision. This is why I commented in some detail again. However, I'm not willing to review it again. It is not the task of a reviewer to improve argumentation, the order of presentation, the completeness of information, and the language details at the level the manuscript is at in this version. This has to be worked out among authors.

**List of issues still open from the review of version 1 (reply by authors, new comment):**

Major issues:

2.        In some parts, the understanding of the underlying physics has to be discussed in more detail. For the first core concept, the information content of the WV channels 8 and 10, the discussion has to be improved at several places throughout the manuscript. Perhaps this method could even be further simplified by skipping the use of the less sensitive channel 8 data. For another central method, the visible channel texture, its limitations (at least for this manuscript's purpose) have to be evaluated and introduced in more detail.

More discussion will be added regarding WV channels, and limitations of visible channel method will be more discussed in the text.
Not really improved. Discussable argumentation was replaced by imprecise argumentation. See below.

3.        Although the argument of convective precipitation information for data assimilation is touched upon in the beginning and shortly mentioned in the end again, the goal of the manuscript stays unclear. For quite some pages, the text seems to present a new, complete, very simple solution for a thunderstorm detection and early warning task. A task on which the satellite community has been working on for quite some 20 years. Major improvement seems to be reachable, because of the new GOES capabilities. Only when it comes to test cases and systematic verification, the limitations become obvious. This is where the simple solution presented would have to become more complex – as many working detection codes are. These limitations have to be discussed in the light of existing detection and nowcasting methods in literature as well as possible integration of the investigated aspects in such tools.

First paragraph in the introduction will be revised to clarify the purpose of the manuscript.

The clarification only clarifies the use of latent heating nudging in models. It still does not explain the aim of this manuscript. The title suggest it is storm detection and not provision of heating fields for data assimilation. This issue comes down to a simple question: What is your paper aiming at? Please give this answer in the introduction.

You do not give a clear aim in the introduction and you do not state whether you reached any in the final conclusions. Your method is too simple to be a detection or warning tool. It is not a full algorithm, but only proposing and testing concepts. In the current state, the paper still does provide neither a closed concept nor an independent validation (you use the month data set to "calibrate" your method). You are missing more than 50% of all convective cell, according to MRMS-C radar data. The quality of your Tb algorithm for early convection is still unclear to me, because of the

limitations of your presentation. Does it add any lead time compared to radar MRMS-C detection? Or is this all not important for you, because you aim for latent heat nudging fields?

4.      The "statistical results" section is not well presented up to now. Definitions and basis are not given clearly. Some of the statements and numbers seem questionable, in part, because of the limitations of the presentation.

The whole section of 'statistical results' will be modified to present one-month results first and then discuss results choosing different thresholds for the methods. Definitions of POD and FAR will be provided in the beginning of this section. The authors agree that providing results using different period of MRMS data in the validation separately can be confusing to readers.
Therefore, in the revised manuscript, only one set of POD and FAR will be presented from the contingency table that is recreated validating results from the reflectance method against 10 minute MRMS data and results from the Tb method against 30 minute (including future 20minute) MRMS data.
Especially this issue is not solved yet. The presentation of section 4 improved, but is still full of vague wording, fuzzy description, and missing definitions. Even a new questionable issue evolved there with the mixing of the two methods into one contingency table.

Specific and minor issues (line numbers are still copied from the version 1 review):

l.20ff: In the light of the many questions left by the presentation of the statistical evaluation/tuning chapter, the numbers here are not useful. Either present some details of the used definitions and scoring basics or leave them out. I do not understand why there is one set of values for two independent methods (major 4).
As mentioned in Major 4, now only one set of POD and FAR is provided for better readability. Since two methods are used to detect convection in different stages and validation dataset includes convection in all stages, results from the two methods are combined.
How can you combine POD and FAR from two independent methods with two independent goals? This is not useful. See comments below.

l.112ff: You state that Channel 2 data is "normalized by solar zenith angle". Please tell us how you do that. This is not a simple or straightforward task. You could normalize reflectivity, but for the texture signal cos(SZA) will not do the trick. The apparent lumpiness increases following a complex dependence on SZA and is strongly dependant on the cloud top structure. (major 2)
Channel 2 data was divided by cos(SZA). A measure of lumpiness used in this study is horizontal gradients of the cloud top surface and thus, shadows (low reflectance) or even brighter surface enhanced by the effect of complexity on SZA actually helps distinguish convective regions from flat cloud top surfaces that are less affected by SZA.
I understand the background idea. But this is not a sufficient normalization, because brightness decreases with decreasing cos(SZA). You are correct, you could compensate for that by dividing by cos(SZA). But for lumpiness it works the other way round: lumpiness/gradient values increase for decreasing cos(SZA)! Lumpiness is at minimum at noontime and strong for low sun!

l.141: "GOES-R CI algorithm". Can you please give a reference?
It is no longer operational products but, it is based on Mecikalski and Bedka (2006) or Mecikalski et al. (2010) cited in this paper.
Give the explanation and reference in the paper not here!

l.155ff: "…updrafts of water vapour…", "…GOES-ABI… can." – You seem to formulate a misconception here. You cannot really see the rising water vapor. The signal is not strong enough. In a WV channel, you do see the water vapor background in midtroposphere. You cannot see low-level dry-convection below condensation level. If you start to see convection cells in this data, it is the cloud body itself you see. Only once the cloud has formed, the emissivity is large enough to dominate the thermal signal in the WV channel. The cloud top "punches through" the background water vapor. Unless the mid-troposhere is very dry, you cannot see what's going on at lower. Please clarify and adjust the discussion here. (major 2)

We agree that these sentences were misleading. "It will be modified to Operational weather radars cannot observe small cloud water, but water vapor absorption bands in GOES-ABI, are more sensitive to these small droplets. During the early convective stages, Tbs that are sensitive to water vapor will decrease due to condensed cloud water droplets aloft generated by a strong updraft." (line 163-165). What is "small cloud water"? Probably droplets. It's not always liquid water! "WV bands are more sensitive to small droplets". Discussable. "During early convective ... Tb WV will decrease ... by strong updraft ..."? You replace the statement that WV updrafts are visible in WV channels, by the statement that strong updrafts create water droplets and are visible in WV channel due to that. Weak updrafts create the same signal in WV, even cirrus anvil clouds create the same signal. Your argumentation is not precise.

L.270: "… most of convective regions align well with high reflectivity regions in Fig. 2c…", You should not only talk about false alarms, but also about the POD. You are missing large regions with coldest temperatures and, thus, a quite obvious signal just next to the region you detected along 43 N and 93 W to 94 W! These regions shows up clearly in a cold absolute 11.2 mu data and in 11.2 lumpiness! This is opposed to your above statement on IR lumpiness and is a large area completely missed by your mature storm detection. Please discuss. (major 3)

It seems like it wasn't clear from two figures being separate and thus, overlaying figure will be added as Figure 5 (with the description of the figure "For a better comparison between detection from GOES and MRMS, convective regions detected by GOES (Fig. 4d) are parallax corrected with a constant cloud top height of 10km and plotted on top of the MRMS map (Fig. 2d), and it is shown in Fig. 5. Most of convective regions align well with high reflectivity regions in Fig. 2c and convective regions in Fig. 2d." in line 276-278). You will see that over 43N and 93W to 94W, convective regions are also detected by the method.

. By showing this overlay, at least, one can see better that for the most western parts of the GOES detection there are convective radar matches. However "Most of convective regions align well …" is not true. Most (= the majority) of the convective radar area does not show overlap with your detections. You can argue (in the manuscript), but you cannot simply deny it!

l.281: "Growing clouds…" Are these boxes result of your method or did you place them by hand as marker to talk about certain areas. You are talking about the purple and blue boxes next. What about yellow and green? Did you miss them? Please make clear. (major 4)

Boxes are clouds that were detected by the Tb method. This sentence will be changed to "Growing clouds shown in purple, blue, yellow, and green boxes are detected by the Tb method, but all starting from different time." to make it clear for readers (line 288-289). Clouds in yellow and green boxes are discussed few sentences after this sentence (line 294 and 295)

Sorry, I missed the description of yellow and green obviously.

Still the problem of this figure and section is that it is purely descriptive and confusing. It still sounds as if you labelled four arbitrary clouds in these two images at two arbitrary points of time. Two clouds were detected early enough, i.e. before MRMS convective flag (purple, yellow), one late (blue), for one the reader can't tell whether detected in time (green) and at least one can be seen for which no Tb detection was issued at all (98W, 38.3 N).  If you want to demonstrate the functioning of your method, you have to think about a different way to show it. Maybe time-step-by-time-step image series with automatic display of Tb detection at exactly the moment they happen.

l.295, Section 4.3, Statistical results: Please start this chapter with a clear definition of the "truth" you compare to, of a hit, miss, false alarm and false positive, all derived skill scores. What is the basic element of your scoring? Is it a grid point, a storm, or a 5x5 window? Please state that for all scores you derive for the early convection as well as the mature convection steps. Right now, this important information is (in part) hidden in the following chapter, but the reader has to guess most of the time. (major 4)

The definition of POD and FAR and its application in this study will be added in the beginning of section 4.3 (line 307-315)

Improved but still not complete. You give this strange mixed contingency table now. It is not useful. You have to do that separately for both your methods. You have to describe all parameters. See next points. What is the basis of your comparison? My question still is: Is it a grid point, a storm, or a 5x5 window.?

l.310: What is the "accuracy" you are talking about? You have to introduce it. It seems to be the correct positives. Please clarify in the beginning of the chapter. (major 4)

It will be clarified "100% accuracy of detecting convection as in MRMS" (line 369).

What is 100% accuracy? This is still not defined. Does that mean 100% of Tb method warnings are MRMS-C cases within 20 min? Or is it 100% of all MRMS-C cases have been detected before. Or is it something else?

l.311: "because most of early convection does not have such a strong updraft". No. It's because it is detected late. See my comment on the WV channels misconception above. In some situations, convection has to reach a considerable height before it can be detected. This is the reason why Mecikalski, Zinner or Guillou did not just use a WV channel to detect early stages. (major 2)

This sentence was misleading as well. It will be changed to "it misses much of the convection and loses an ability to detect convection earlier than radar because not all convective clouds have such a strong updraft." (line 369).

As before, wording is still imprecise. The strength of the updraft (the vertical wind) can not be seen in the WV channels. It can be seen only (in cloud top cooling), if the cloud top is high enough. If you agree, you have to write this. If not, you have to convince me.

l.333f: "improvements in both FAR and POD (lower FAR and higher POD) when later data are included." This is not surprising and it is just tuning values. It would improve further, if you would include another 10 minutes, or even -10 minutes. Unless you can tell us a very good reason resulting from the function of the MRMS algorithm, I would suggest not showing the alternative numbers. They are not much different anyway. (major 4)

The whole section of 'statistical results' is modified to reflect this comment.

This problem still exists unchanged.

**Excuse me … some new issues (line numbers from new version 2 now):**

l.220: Please state which VIS channel!

Section 4.3: You need to split the analysis into section "4.3.1 Mature convection detection method" and "4.3.2 Early convection method". It makes no sense to mix up the results as you do now.

l. 306: What is both methods? You should give unique names to the methods and use them throughout the manuscript. GOES-C seems to be the VIS maturity detection. You do not state this. Do these numbers refer to pixel? To area? To objects? Not clear. And I asked this in the review 1.

l.320: Before this line it should be section headline "4.3.1 Mature…". And here you state yourself: it makes no sense to mix up results of both methods.

l.322: "(reflectance at channel 2 … cloud top surfaces)" - This all has to be defined and fixed in section 3.2. Otherwise you have to make it clear there that you intend to tune all these parameters here.

l. 337 ff: You should not show this test. Your maturity detection is aimed at detecting mature convection at the time t of detection. Obviously you include MRMS-C cases of the time "t+10min" IN ADDITION to MRMS-C cases at time "t", right? Otherwise not all the numbers would improve.
If this is the case, this is just number tuning. If you would include time "t+5min" and "t+15" and so on, it would always improve numbers without any improvement of quality of the mature convection detection. And I requested not to do this in review 1.

l.355: Here is the point for headline "4.3.2 Early …".

l.360: "27971 and 73204" This means the basis of your analysis is everything your method detects as convection? With which thresholds? How do you get these "windows"? If this is right you can not derive a POD, because you do not analyse an independent truth. Please state this in the manuscript.

l.366: What about the +30 min you just mentioned?

l. 368: "Accuracy" is still not defined.

l.372: "Growth rate observed": You rather mean "Cooling rate observed". Similar in line 375.

L385: "14.4% is achieved, and 96.4% of false alarm pixels". Can the reader check this statement anywhere in the presented results? Please tell him in the manuscript.

Fig. 2: Do (c) and (d) really show the same point in time as stated? The two precipitation fields (d) and the derived product (d) do not look like! The precipitation areas north of 44 N have hardly any matching feature.

Fig. 5: Suddenly it's 2230 UTC  and not 1930 UTC as before? Something wrong with the time and date?

Fig. 7: Please give all parameters kept constant in the caption or image. And please extend caption with some information on method discussed. Reflectance method for mature convection detection.

Fig. 8: Skip figure b. And again. Extend caption as before.

Fig. 9: "when the pixel was" ...Should better read "…if a pixel was assigned to be convective by MRMS, but not detected by method XXX …" Correct?

Tab. 1: Please extend caption again. This is too short to be understood.

---

## Referee Report (RR3)

The authors did a good job responding to my previous comments. This article is recommended for publication after replying to the following minor comments.

1. Line 35, "... add heating as frequent as possible, …" : why it should be "frequent"? or replace by "as accurately as possible".
2. Line 86 : add brackets to "Bedka et al. 2016". Same in line 314 for Vicente et al. 2002 and please check citation styles throughout the manuscript.
3. Line 101 : change "... temporal trends of the data were used but, since …" to "... temporal trends of the data were used, but since …"
4. Line 134 : change "... as they progress upwards, …" to "as clouds develop upwards"
5. Line 166 : replace "... the magnitude of the gradients are …" by "... the magnitude of the gradients is…"
6. Line 166, "... use horizontal gradients" :  should it be both horizontal and vertical? or remove it and please check throughout the manuscript.
7. Line 216-218, "However, not all the detection by the method is done early since MRMS product is created not just using high reflectivity, it is rather good at detecting early convection." : need correction here for better understanding.
8. Line 269, "... due to the IR's 2km resolution …" :  suggestion "due to their relatively lower spatial resolution"
9. Line 278 : would leave this sentence out.
10. Line 284 : add comma after "... vertical level". Same in line 317 after "...convective regions"
11. Line 291 : suggest to add "together" after "... since both clouds were detected"
12. Line 280 and 325 : please add the names of the cities on the map.
13. Line 327, "during 22:30UTC~22:40UTC" : please use a hyphen. Same in line 337
14. Line 350 : replace by  "... decrease with latitude"
15. Line 348-350 : The effects of solar zenith angle or lower spatial resolution seem to conflict with each other. For this case (greater convective area), then is this probably due to the large SZA as it is in the afternoon? Please clarify.
16. Line 359-364: The information should be also in the caption of Table 3, and please mention how the values in percentage are calculated. The numbers can be provided here together.
17. Line 372 : replace "Most of the detection is …" by "Most of the detection are …"
18. Line 374 : Tables 1 and 2 don't have FAR information.
19. Line 374-375, "Relatively small FAR compared to Tables 1 and 2 would be because Tables 1 and 2 are obtained based on each cloud while FAR and POD are calculated based on each grid point." :  Please give more explanation on this.
20. Line 380 : replace "... which is essentially …" by "... which are essentially …"
21. Line 385-386 : what is meant by "a random chance"? Please state explicitly.
22. Line 395 : replace "It is better to not …" by "It is better not to …". Please clarify more on ".. give any information".
23. Line 427 : suggest to add "in a more effective way" after "... facilitate cloud tracking". Please revise this sentence "... helps the accuracy of the detection method when calculating decreases in Tb of the same cloud."
24. Line 441 : where does the figure "~85%" come from?
25. Line 443-444 : There are already some studies using machine learning algorithms and even deep learning for detecting convective initiation or overshooting cloud tops. Related studies can be mentioned in the Introduction.

26. Figure 2 : In the 4th box, replace "... at channel 8 and 10 are calculated." by "... at channel 8 and 10 is calculated."
27. Figure 7, Line 705: replace "Times next to each box represents …" by "Times next to each box represent …". Should "the mature cloud detection method" be "the growing cloud detection method"?

---

## Author Response (AR2)

The authors appreciate two reviewers' valuable comments once again.

**Response to Reviewer #1**

Main changes in figure and table
- Figure 2 is added.
- Figure 7 is added.
- Figure 2c is modified.
- The order of tables is changed and tables 1 and 2 have updated values using different one-month dataset based on Reviewer #1's comment. Table 2 and 3 in the previous manuscript used to show the number of convective windows including previous time period (e.g. "convective within 10min" would be the sum of the number for "convective at 10-min detection period" and the number for "convective within 10-min after the detection period"), but Table 1 and 2 in the revised manuscript only shows the number for each 10-min period. And additional column is created to show overall accuracy, which is the sum of the 3,4,5$^{th}$ columns divided by the sum of 2,3,4,5$^{th}$ columns. Note that the numbers are different from previous manuscript because different data are used.

Major issue 2 and minor issues regarding limitation of visible channel method:
Limitation of visible channel method (mature cloud detection method in the revised manuscript) is more discussed more broadly in the revised manuscript. The dependency on SZA is discussed in line 345.

Major issue 3:
The clarification only clarifies the use of latent heating nudging in models. It still does not explain the aim of this manuscript. The title suggest it is storm detection and not provision of heating fields for data assimilation. This issue comes down to a simple question: What is your paper aiming at? Please give this answer in the introduction.
You do not give a clear aim in the introduction and you do not state whether you reached any in the final conclusions. Your method is too simple to be a detection or warning tool. It is not a full algorithm, but only proposing and testing concepts. In the current state, the paper still does provide neither a closed concept nor an independent validation (you use the month data set to "calibrate" your method). You are missing more than 50% of all convective cell, according to MRMS-C radar data. The quality of your Tb algorithm for early convection is still unclear to me, because of the limitations of your presentation. Does it add any lead time compared to radar MRMS-C detection? Or is this all not important for you, because you aim for latent heat nudging fields?
We now discuss the ultimate goal of this work, which is to identify convection for use in weather forecast model initialization. This matters because it skews the algorithm towards avoiding false negatives (i.e. assigning convection where there is none) while not penalizing the algorithm excessively for missing convection. The authors agree that this is a testing concept and it might need more modification to be used in operation and this is now clearly stated in lines 93-94 and 433-435.

The authors agree that it might have sounded as if the same month data were used to calibrate the actively growing cloud, but actively growing cloud method calibration also included data during nighttime hours while results for combined methods are only evaluated data during day since the mature convective cloud detection method uses reflectance data. Since it can cause confusion, different one-month data are used to provide a reason for choosing the two thresholds for channel 8 and 10.

Even though the two tables (table 2 and 3) were given to show the ability of the method to detect convection earlier than MRMS, there wasn't a clear lead time compared to the MRMS product. Lines 214-215 are added.

Major issue 4 and minor issues regarding combining two methods:
The two methods were combined because precipitation flags from MRMS do not separate convective clouds in different stages, but POD and FAR for each of the two methods are added along with the combined POD and FAR in line 369-370.

l.141: "GOES-R CI algorithm". Can you please give a reference?
Since it's not operational anymore it is replaced with previous studies.

l.155ff: "…updrafts of water vapour…", "…GOES-ABI… can." – You seem to formulate a misconception here. You cannot really see the rising water vapor. The signal is not strong enough. In a WV channel, you do see the water vapor background in midtroposphere. You cannot see low-level dry-convection below condensation level. If you start to see convection cells in this data, it is the cloud body itself you see. Only once the cloud has formed, the emissivity is large enough to dominate the thermal signal in the WV channel. The cloud top "punches through" the background water vapor. Unless the mid-troposhere is very dry, you cannot see what's going on at lower. Please clarify and adjust the discussion here. (major 2)
We agree that these sentences were misleading. "It will be modified to Operational weather radars cannot observe small cloud water, but water vapor absorption bands in GOES-ABI, are more sensitive to these small droplets. During the early convective stages, Tbs that are sensitive to water vapor will decrease due to condensed cloud water droplets aloft generated by a strong updraft." (line 163-165).
What is "small cloud water"? Probably droplets. It's not always liquid water! "WV bands are more sensitive to small droplets". Discussable. "During early convective ... Tb WV will decrease ... by strong updraft ..."? You replace the statement that WV updrafts are visible in WV channels, by the statement that strong updrafts create water droplets and are visible in WV channel due to that. Weak updrafts create the same signal in WV, even cirrus anvil clouds create the same signal. Your argumentation is not precise.

"Small cloud water" is changed to "small hydrometeor" and the corresponding sentence is modified in line X. "Strong updrafts" was used as strong updraft is what distinguishes convective core from cirrus or anvil clouds, and the method uses large $T_b$ decrease to detect convection. We do not think that large decrease in $T_b$ is observed in cirrus or anvil clouds. One example is shown below. It's the same scene from Fig. 4a and 4b but the anvil cloud over 93W is zoomed in and color bar is set to observe Tb decrease more carefully. Note that the color bar is set to 210K-220K which is 10K difference between two time steps. Lower figure is the zoomed in figure of Fig. 5d. Anvil clouds over blue box regions or around the blue box regions didn't really show change in $T_b$, while some clouds show larger decrease (still less than 10K), and these clouds are detected by the mature cloud detection method.

[Figure]

By showing this overlay, at least, one can see better that for the most western parts of the GOES detection there are convective radar matches. However "Most of convective regions align well …" is not true. Most (= the majority) of the convective radar area does not show overlap with your detections. You can argue (in the manuscript), but you cannot simply deny it!
We have changed the text to properly interpret the scene. It now reads that the convective regions while not perfectly aligned due to a number of dynamic geometric reasons, do have a high degree of correspondence between the two detection methods.

Sorry, I missed the description of yellow and green obviously.
Still the problem of this figure and section is that it is purely descriptive and confusing. It still sounds as if you labelled four arbitrary clouds in these two images at two arbitrary points of

time. Two clouds were detected early enough, i.e. before MRMS convective flag (purple, yellow), one late (blue), for one the reader can't tell whether detected in time (green) and at least one can be seen for which no Tb detection was issued at all (98W, 38.3 N). If you want to demonstrate the functioning of your method, you have to think about a different way to show it. Maybe time-step-by-time-step image series with automatic display of Tb detection at exactly the moment they happen.

Times of detection by GOES and MRMS are added in the figure.

Improved but still not complete. You give this strange mixed contingency table now. It is not useful. You have to do that separately for both your methods. You have to describe all parameters. See next points. What is the basis of your comparison? My question still is: Is it a grid point, a storm, or a 5x5 window.?

We have tried to clarify by providing FAR and POD for each method as suggested by the reviewer, along with the combined results. These are calculated based on MRMS's grid as mentioned in lines 350 and 354.

What is 100% accuracy? This is still not defined. Does that mean 100% of Tb method warnings are MRMS-C cases within 20 min? Or is it 100% of all MRMS-C cases have been detected before. Or is it something else?

We have changed the explanation for the table in lines 203-217 to make this cleaer.

l.311: "because most of early convection does not have such a strong updraft". No. It's because it is detected late. See my comment on the WV channels misconception above. In some situations, convection has to reach a considerable height before it can be detected. This is the reason why Mecikalski, Zinner or Guillou did not just use a WV channel to detect early stages. (major 2)

This sentence was misleading as well. It will be changed to "it misses much of the convection and loses an ability to detect convection earlier than radar because not all convective clouds have such a strong updraft." (line 369).

As before, wording is still imprecise. The strength of the updraft (the vertical wind) can not be seen in the WV channels. It can be seen only (in cloud top cooling), if the cloud top is high enough. If you agree, you have to write this. If not, you have to convince me.

This sentence is changed in line 220-221.

l.333f: "improvements in both FAR and POD (lower FAR and higher POD) when later data are included." This is not surprising and it is just tuning values. It would improve further, if you would include another 10 minutes, or even -10 minutes. Unless you can tell us a very good reason resulting from the function of the MRMS algorithm, I would suggest not showing the alternative numbers. They are not much different anyway. (major 4)

The whole section of 'statistical results' is modified to reflect this comment.

This problem still exists unchanged.

l.220: Please state which VIS channel!

The section is modified so the VIS channel is mentioned before.

Section 4.3: You need to split the analysis into section "4.3.1 Mature convection detection method" and "4.3.2 Early convection method". It makes no sense to mix up the results as you do now.

Part related to the early convection method is moved to methodology section, and FAR and POD for each method are provided separately.

l. 306: What is both methods? You should give unique names to the methods and use them throughout the manuscript. GOES-C seems to be the VIS maturity detection. You do not state this. Do these numbers refer to pixel? To area? To objects? Not clear. And I asked this in the review 1.

The name for the methods are given as growing cloud detection method and mature cloud detection method. GOES-C stands for GOES convective (results combining the two methods). Line 352-354 is added to explain the numbers.

l.322: "(reflectance at channel 2 … cloud top surfaces)" - This all has to be defined and fixed in section 3.2. Otherwise you have to make it clear there that you intend to tune all these parameters here.

It was defined in line 257 and 238 in section 3.2 of the previous manuscript.

l. 337 ff: You should not show this test. Your maturity detection is aimed at detecting mature convection at the time t of detection. Obviously you include MRMS-C cases of the time "t+10min" IN ADDITION to MRMS-C cases at time "t", right? Otherwise not all the numbers would improve.
If this is the case, this is just number tuning. If you would include time "t+5min" and "t+15" and so on, it would always improve numbers without any improvement of quality of the mature convection detection. And I requested not to do this in review 1.

Figure 8b is removed.

l.360: "27971 and 73204" This means the basis of your analysis is everything your method detects as convection? With which thresholds? How do you get these "windows"? If this is right you can not derive a POD, because you do not analyse an independent truth. Please state this in the manuscript.

It wasn't explained well enough and thus it is modified.

l.366: What about the +30 min you just mentioned?

30 minutes meant including next 20-minute after the detection period of 10min. But since it sounds confusing, it's changed.

l. 368: "Accuracy" is still not defined.

It is modified.

l.372: "Growth rate observed": You rather mean "Cooling rate observed". Similar in line 375.

They are modified.

L385: "14.4% is achieved, and 96.4% of false alarm pixels". Can the reader check this statement anywhere in the presented results? Please tell him in the manuscript.

96.4% was mentioned as an additional information after analysis.

Fig. 2: Do (c) and (d) really show the same point in time as stated? The two precipitation fields (d) and the derived product (d) do not look like! The precipitation areas north of 44 N have hardly any matching feature.

Fig. 2c is replaced with the MRMS product so that it matches with Fig. 2d.

Fig. 5: Suddenly it's 2230 UTC and not 1930 UTC as before? Something wrong with the time and date?

Sorry for the typo. It's been changed.

Fig. 7: Please give all parameters kept constant in the caption or image. And please extend caption with some information on method discussed. Reflectance method for mature convection detection.

They were added.

Fig. 8: Skip figure b. And again. Extend caption as before.

Fig. 8b was removed and the extended caption is added.

Fig. 9: "when the pixel was" ...Should better read "...if a pixel was assigned to be convective by MRMS, but not detected by method XXX ..." Correct?

It was changed.

Tab. 1: Please extend caption again. This is too short to be understood.

It is extended.

**Response to Reviewer #2**

Line 7: As a suggestion, "The ability to detect convective regions and to add latent heating to drive convection in weather forecast models is the most important skill in forecasting severe weather systems."
It's been changed in line 7.

Line 12-15: As a suggestion, "Relatively new geostationary satellites, Geostationary Operational Environmental Satellites-16 and -17 (GOES-16 and GOES-17), along with Himawari-8, can make up for this lack of vertical information through the use of very high spatial and temporal resolutions, allowing to better observe bubbling features on convective cloud tops."
It's been changed in line 13.

Line 20: Please provide a long name of MRMS.
It's added in line 21.

Line 21: Please specify accuracy measures reported here.
It's been changed in line 22.

Line 46: The 'peakedness' and 'surrounding area' criteria are not well explained by the following sentence.
More explanation is added in line 48-50.

Line 50: Please replace "at -10°C or higher" to "at -10°C height or above"
Its' been changed in line 52.

Line 50-51: Does Zhang and Qi (2010)'s method use the threshold for convective precipitation?
6.5kg/m2 was their threshold. But since it wasn't explained well enough, the phrase has been changed in line 52.

Line 62: Tb has not yet been defined.
It was defined in line 64.

Line 87: Please specify which Meteosat series.
It's been changed in line 90.

Line 93: Please explain how "mesoscale sectors" is defined.
Explanation is added in line 97.

Line 96 "errors from cloud movements": Could you please elaborate on the errors from cloud

movements?
Explanation is added in line 100-104.
Line 99: Change "Tb" to "Tb from IR channels"
It's been changed in line 105.

Line 140-142 "It is a rather sophisticated classification...": Shouldn't this sentence be moved after the sentence "Details of the classification can be found in Zhang et al. (2016)." in line 137.
It's been moved to line 143.

Line 182 "inverse Gaussian": It might be misleading with the term, inverse Gaussian distribution used in probability theory. Please use a different term.
It's been changed to "upside down Gaussian".

Line 183: Please change "Tb shape" to "the Tb matrix"
It's been changed.

Line 190-191 "Since one-minute data can be noisy, the decreasing trend was considered instead of an actual difference in Tb during 10minutes.": would leave this sentence out.
It's been removed.

Line 199 "Using two channels help find the same clouds in different levels.": How are both channels used to find the same clouds? Does it mean that both channels need to satisfy the conditions? Please clarify, and change "help" to "helps".
Phrase is modified. The fact that both channels don't need to satisfy the conditions is mentioned in line 97-98.

Line 202: Change "make" to "makes"
It's been changed.

Line 221: Change "relative to" to "due to"
It's been changed.

Line 223: Please correct "texturs".
It's been changed.

Line 236: It would be better to put the paragraph for the Sobel operator after the paragraph that describes screening scenes with VIS and IR channels (i.e., lines 223-234).
It's been changed.

Line 238: Please mention how the thresholds, 0.4 and 0.9, were obtained. Change "implies" to "imply"
It was mentioned in the text that results using different thresholds are compared in section 4.3.

Line 239: "... with very high gradients." to "... with very high gradients, respectively."
It's been changed.

3. Methodology: It would be good to have a flowchart for the methodology so that readers can have an overview of the methods used in this study.
Flowchart for the growing cloud detection method is added in Figure 2.

Line 246: Isn't it southwest, not southeast?
It's been changed.

Line 248-249: The radar data could be described in section 2.2, and the sentence would be something like "The two cells appeared in the composited NEXRAD radar data…"
Fig. 2c is changed to MRMS SeamlessHSR product.

Line 254: "channels 10" to "channels 10 (7.3 μm)". Please correct "the white circle".
It's been changed.

Line 259-260: Change "The only two matrices in this scene that satisfied both criteria of maintaining the shape of developing cells and growing vertically over ten time steps were the two in blue circles." to "The two matrices in the blue box satisfied both criteria of maintaining the shape of developing cells and growing vertically over ten time steps.", or please revise it better.
It's been changed in line 291-292.

Line 276: Please describe what parallax correction is and how it is calculated/corrected.
Description is added in line 309-313.

Line 289 "from different time": It would be better to provide the times when each coloured box was detected.
Times of GOES and MRMS detection for each box are added in Figure 6 and text corresponding to Figure 6 is modified in line 322-335.

Line 291: "minute" to "minutes"
It's been changed.

Line 293 "This shows a need to use both channels in the detection.": Does it mean that both channels need to be satisfied together in the detection? Please clarify.
The phrase is modified in line 334-335.

Line 296 "the Tb method": It would be good to name the two methods for growing cloud detection and mature cloud detection method, respectively, in the Methodology section beforehand and use them in the following discussions.
It's been changed throughout the manuscript.

Line 300 "Black regions superimposed on the brightness temperature map in Fig. 6c represent…": It would be good to also superimpose the detected mature convective clouds on the MRMS map, same as in Figure 5 for a better comparison.
The figure is added as Figure 7.

4.3 Statistical results with one-month data: The part that explains the method seems to be quite huge, so it might be better to move them to the Methodology section.
The part where thresholds for the growing cloud detection method are discussed is moved to methodology section, but the part where thresholds for the mature cloud detection method are discussed stays in this section because it uses FAR and POD.

Line 329: Using a threshold of 0.5 looks a reasonable compromise as well. It would be good to discuss the optimal balance between POD and FAR. It would be good to explain which one is the more important factor and why.
Few sentences are added in line 380-383.

Line 333 "... mature convective clouds in the earlier stage…": Does it mean detecting clouds that grow into mature convective clouds?
This sentence is modified in line 386.

Line 335-336 "... for its potential use in the short-term forecast, …": Is it okay to have small POD in the short-term forecast. Please give an explanation on it.
More explanation is added in line 391-392.

Figure 7: Please remove "(a)".
It's removed.

Line 337-338 "Figure 8b shows results including MRMS data 10 minutes after the detection period.": What does it mean to include MRMS data 10 min. after the detection period? Why were the data 10 min. after the detection period included? Please clarify.
From the decreased FAR and increased POD, it can be implied that false alarms that were detected by GOES but not by MRMS during the detection period were actually convective clouds defined by MRMS at later time. However, authors agree that it can be confusing to readers therefore, Figure 8b is removed.

Line 339-340 "Fig. 8b still shows its ability to detect convection earlier than MRMS.": It is unclear how Fig. 8b shows the ability of detecting convection earlier than MRMS.
Figure 8b is removed.

Figure 9: Please add labels for x-axis with units.
Labels are added in the figure.

Line 360: When all the windows were first collected, what were the thresholds used for each channel?

It was added in line 204.

Line 365: Please add "... for channel 8 and 10, respectively." at the end.
It was added.

Table 2 and 3: How are the first two columns obtained? How is the overall accuracy calculated? Please more clarify how the experiment is conducted for Table 2 and 3.
Clarification sentence is added in line 209-214.

Line 375-376 "Therefore, it makes sense again that the growth rate at channel 10 has to be bigger to catch up lower Tb in channel 8.": Why does the growth rate at channel 8 needs to be caught by channel 10?
Growth rate at channel 8 doesn't need to be caught by channel 10. This sentence was to explain in a mathematical sense that since channel 8 usually has lower Tb than channel 10 because it sees high-level water vapor and as clouds grow, those channels exhibit similar Tb, Tb change over time at channel 8 should be larger than Tb change in channel 10. But we understand that this sentence can confuse the readers and therefore, this sentence is removed.

Line 377-378: Please give a reference.
This was just authors' opinion on why clouds don't precipitate even with rapid growth.

Line 382-383: Could you explain more why false alarms are most detrimental to data assimilation? and please give a reference.
The sentence is added in line 391-392.

Line 385-387: What if the POD is so small that only a few convective clouds with most precipitation are detected? Could you please explain more on the last two sentences?
Lines 410-419 are changed.

Line 390: "...to detect convective clouds" to "...to detect convective clouds in two different stages…"
It's been changed.

Line 403 "extremely": It sounds a bit too strong. Please remove it or replace it with another word.
It's been removed.

Results and Discussion: The labels in most figures are so small that can hardly be identified.
Labels in the figures are enlarged.

Please check typos and tense throughout the manuscript.
Sorry for the typos. They were checked.

---

## Author Response (AR3)

**The authors greatly appreciate the reviewer for taking his/her time to review this manuscript and all the invaluable comments. Simple modifications (e.g. typo) are in red color and other changes are in green color in the response as well as in the revised manuscript.**

1. Line 35, "... add heating as frequent as possible, …" : why it should be "frequent"? or replace by "as accurately as possible".
It's modified in line 35.

2. Line 86 : add brackets to "Bedka et al. 2016". Same in line 314 for Vicente et al. 2002 and please check citation styles throughout the manuscript.
They are changed in lines 88 and 316 as well as line 93

3. Line 101 : change "... temporal trends of the data were used but, since …" to "...temporal trends of the data were used, but since …"
It's modified in line 103.

4. Line 134 : change "... as they progress upwards, …" to "as clouds develop upwards"
It's modified in line 136.

5. Line 166 : replace "... the magnitude of the gradients are …" by "... the magnitude of the gradients is…"
It's modified in line 168.

6. Line 166, "... use horizontal gradients" : should it be both horizontal and vertical? Or remove it and please check throughout the manuscript.
It should only be horizontal gradient, therefore it's modified to "horizontal gradient" in line 168.

7. Line 216-218, "However, not all the detection by the method is done early since MRMS product is created not just using high reflectivity, it is rather good at detecting early convection." : need correction here for better understanding.
This sentence is modified to "However, not all the detection by the method is done early since MRMS can also sometimes assign early convection as convective before it produces high reflectivity." in lines 218-220.

8. Line 269, "... due to the IR's 2km resolution …" : suggestion "due to their relatively lower spatial resolution"
It's modified in line 270.

9. Line 278 : would leave this sentence out.
This sentence is removed.

10. Line 284 : add comma after "... vertical level". Same in line 317 after "...convective regions"
These are modified in line 286 and 319.

11. Line 291 : suggest to add "together" after "... since both clouds were detected"
It's modified in line 293.

12. Line 280 and 325 : please add the names of the cities on the map.
Names of states are added in Figure 3 and 7.

13. Line 327, "during 22:30UTC~22:40UTC" : please use a hyphen. Same in line 337
They are modified in lines 328-330 and 339.

14. Line 350 : replace by "... decrease with latitude"
The whole lines 350-354 are modified based on comment 15.

15. Line 348-350 : The effects of solar zenith angle or lower spatial resolution seem to conflict with each other. For this case (greater convective area), then is this probably due to the large SZA as it is in the afternoon? Please clarify.
These sentences are modified to "This could be due to dependency of lumpiness on some geometrical considerations. Lumpiness is a function of the pixel spatial resolution, differences in optical depth and shadows. Spatial resolution decreases away from the equator, but higher solar zenith angles (due to altitude or time of day) not only increases optical depth, they also increase shadows. While this can of course be dealt with, it was ignored in this study which serves primarily as a proof of concept, as the method generally finds convective core correctly." in lines 350-354.

16. Line 359-364: The information should be also in the caption of Table 3, and please mention how the values in percentage are calculated. The numbers can be provided here together.
Caption of Table 3 is modified and the actual numbers to calculate the percentages are added in Table 3.

17. Line 372 : replace "Most of the detection is ..." by "Most of the detection are ..."
It's modified in line 375.

18. Line 374 : Tables 1 and 2 don't have FAR information.
This sentence is revised based on comment 19.

19. Line 374-375, "Relatively small FAR compared to Tables 1 and 2 would be because Tables 1 and 2 are obtained based on each cloud while FAR and POD are calculated based on each grid point." : Please give more explanation on this.
This sentence is modified to "FAR in Tables 1 and 2 (1- overall accuracy values)" in lines 377-378.

20. Line 380 : replace "... which is essentially ..." by "... which are essentially ..."
It's modified in line 383.

21. Line 385-386 : what is meant by "a random chance"? Please state explicitly.
It's modified in line 389.

22. Line 395 : replace "It is better to not …" by "It is better not to …". Please clarify more on ".. give any information".
This sentence is modified to "In data assimilation, it is preferable to provide no constraints than to provide the model with the incorrect location of convection." In lines 399-400.

23. Line 427 : suggest to add "in a more effective way" after "… facilitate cloud tracking". Please revise this sentence "… helps the accuracy of the detection method when calculating decreases in Tb of the same cloud."
"In a more effective way" is added in 431 and the sentence is changed to "helps reduce uncertainties coming from cloud tracking when calculating decreases in $T_b$ of the same cloud" in lines 431-432.

24. Line 441 : where does the figure "~85%" come from?
The parenthesis "(100%-FAR of 14.4%)" is added in lines 446-447.

25. Line 443-444 : There are already some studies using machine learning algorithms and even deep learning for detecting convective initiation or overshooting cloud tops. Related studies can be mentioned in the Introduction.
"With the recent growing interest in machine learning techniques, many studies have applied machine learning methods in detecting convection (Han et al., 2019; Zhang et al., 2019; Cintineo et al., 2020), but knowledge in physical features of convective clouds is still required to construct a model that correctly learns during training." is added in lines 64-67 with three new references.

26. Figure 2 : In the 4th box, replace "… at channel 8 and 10 are calculated." by "… at channel 8 and 10 is calculated."
Figure 2 is updated.

27. Figure 7, Line 705: replace "Times next to each box represents …" by "Times next to each box represent …". Should "the mature cloud detection method" be "the growing cloud detection method"?
The caption of Figure 7 is modified in lines 709-710 based on these comments.